# Differentially Private Decoupled Graph Convolutions for Multigranular Topology Protection

**Eli Chien**[*]
UIUC & GaTech
ichien3@illinois.edu
ichien6@gatech.edu

**Wei-Ning Chen**[*]
Stanford University
wnchen@stanford.edu

**Chao Pan**[*]
UIUC
chaopan2@illinois.edu

**Pan Li**
GaTech
panli@gatech.edu

**Ayfer Özgür**
Stanford University
aozgur@stanford.edu

**Olgica Milenkovic**
UIUC
milenkov@illinois.edu

## Abstract

Graph Neural Networks (GNNs) have proven to be highly effective in solving real-world learning problems that involve graph-structured data. However, GNNs can also inadvertently expose sensitive user information and interactions through their model predictions. To address these privacy concerns, Differential Privacy (DP) protocols are employed to control the trade-off between provable privacy protection and model utility. Applying standard DP approaches to GNNs directly is not advisable due to two main reasons. First, the prediction of node labels, which relies on neighboring node attributes through graph convolutions, can lead to privacy leakage. Second, in practical applications, the privacy requirements for node attributes and graph topology may differ. In the latter setting, existing DP-GNN models fail to provide multigranular trade-offs between graph topology privacy, node attribute privacy, and GNN utility. To address both limitations, we propose a new framework termed Graph Differential Privacy (GDP), specifically tailored to graph learning. GDP ensures both provably private model parameters as well as private predictions. Additionally, we describe a novel unified notion of graph dataset adjacency to analyze the properties of GDP for different levels of graph topology privacy. Our findings reveal that DP-GNNs, which rely on graph convolutions, not only fail to meet the requirements for multigranular graph topology privacy but also necessitate the injection of DP noise that scales at least linearly with the maximum node degree. In contrast, our proposed Differentially Private Decoupled Graph Convolutions (DPDGCs) represent a more flexible and efficient alternative to graph convolutions that still provides the necessary guarantees of GDP. To validate our approach, we conducted extensive experiments on seven node classification benchmarking and illustrative synthetic datasets. The results demonstrate that DPDGCs significantly outperform existing DP-GNNs in terms of privacy-utility trade-offs. Our code is publicly available[2].

## 1 Introduction

Graph learning methods, such as Graph Neural Networks (GNNs) [1–5], are indispensable learning tools due to the ubiquity of graph-structured data and their importance in solving real-world problems

---

[*]Equal contribution.
[2]https://github.com/thupchnsky/dp-gnn

37th Conference on Neural Information Processing Systems (NeurIPS 2023).

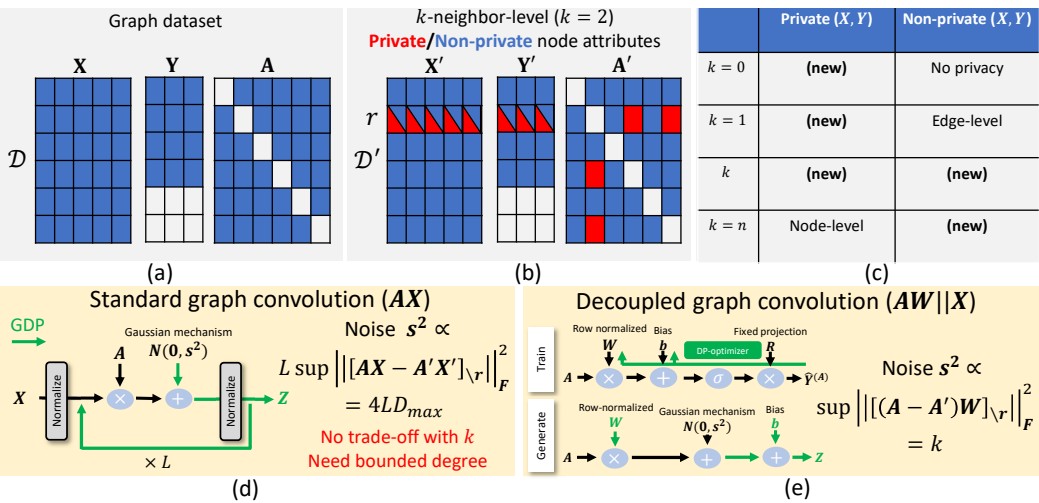

Figure 1: Top: (a) Illustration of a training graph dataset. In the example, the graph involves 6 nodes and does not contain self-loops. Nodes 5 and 6 are left unlabeled in the training dataset $\mathcal{D}$. (b) An illustration of our novel notion of $k$-neighbor-level graph dataset adjacency, with/without node attributes privacy. Red colors indicate entries that are to be replaced in the adjacent dataset $\mathcal{D}'$ with respect to a node $r$. (c) All possible combinations of graph topology and node attribute privacy requirements under our $k$-neighbor definition. Bold letters indicate settings not covered by prior literature. Bottom: Illustration of a (d) standard graph convolution and (e) our decoupled graph convolution design. For decoupled graph convolution, we concatenate the node embedding $\mathbf{Z}$ with node feature $\mathbf{X}$ to obtain the final prediction. See Figure 2 for a more detailed description of the DPDGC model. Note that the required noise for standard graph convolution is independent of $k$, and hence cannot leverage the intrinsic privacy-utility trade-off in $k$-neighbor level adjacency.

arising in recommendation systems [6], bioinformatics [7] and fraud detection [8]. Graph datasets, which typically comprise records of users (i.e., node attributes) and their interaction patterns (i.e., graph topology), often contain sensitive information. For example, financial graph datasets contain sensitive financial records and transfer logs between accounts [9, 10]. Online review system graphs comprise information about customer identities and co-purchase records [11].

Given the sensitive nature of graph datasets, it is paramount to ensure that learned models do not reveal information about user attributes or interactions. Unprotected learning models can inadvertently leak information about the training data even if the data itself is not disclosed to the public [12]. Differential privacy (DP) has been used as *the* gold standard for rigorous quantification of "privacy leakage" of a learning algorithm, as it ensures that the output of a training algorithm remains indistinguishable from that of "adjacent" datasets [13]. Classical DP approaches focus on DP guarantees for model weights (i.e., DP-SGD [14]) when only node attributes are present. This suffices to ensure DP of model predictions for standard classification settings when no graph structure is available. In such settings, the prediction of a user is merely a function of model weights and its own attributes. This assumption does not hold for graph learning methods in general, since graph convolution, by coupling node attributes and topology information, also leverages information from neighboring users at the prediction stage. Furthermore, in practice, there are usually different privacy level requirements for node attributes and graph topology. For example, customer identities can be more sensitive compared to co-purchase records in an online review system. A fine-grained trade-off between graph topology privacy, node attribute privacy, and GNN utility is a necessary consideration overlooked by prior literature. These require rethinking how to achieve adequate DP for GNNs.

**Our contributions.** We perform a formal analysis of Graph Differential Privacy (GDP), ensuring that both the GNN weights and node predictions are DP. The key idea of GDP is to protect the privacy of all nodes at the prediction step except those nodes whose labels are to be predicted, as users who want to know their own predictions must have access to their own data. While this requirement sounds straightforward, it leads to challenges in the analysis due to the interaction between the graph structure and node attributes. To account for different degrees of graph topology and node attribute privacy, we introduce the notion of $k$-neighbor-level adjacency which generalizes the notion of graph dataset adjacency (see Figure 1 (a-c)). It not only unifies previous edge- and node-level adjacency

definitions [15, 16] but also allows for different granularities of privacy protection for the graph topology. Our notion of $k$-neighbor-level adjacency can be used to establish the trade-offs among graph topology privacy, node attribute privacy, and model utility.

The GDP analysis to follow demonstrates that the standard graph convolution operation has two fundamental drawbacks. First, it can be shown that the required DP noise level for a standard graph convolution *does not* decay even when there are no privacy constraints on the graph topology. Hence, the standard graph convolution design fails to exploit the fine-grained trade-off pertaining to graph topology privacy. Second, the required DP noise variance for standard graph convolutions grows at least linearly with the maximum node degree [17], which leads to suboptimal utility. To mitigate these drawbacks, we propose the Differentially Private Decoupled Graph Convolution (DPDGC) design that provably enables the aforementioned trade-off and makes DP noise variance *independent* of the maximum node degree. Our key idea is to prevent direct neighborhood aggregation of (potentially transformed) node features so that the GDP guarantee can be improved via the DP composition theorem (see Figure 1 (e)). This insight sheds new light on the benefits of decoupled graph convolutions, and may lead to further advances in GDP-aware designs. We conclude by demonstrating excellent privacy-utility trade-offs of DPDGC for different GDP settings via extensive experiments on seven node classification benchmarking datasets and synthetic datasets generated using the contextual stochastic block model (cSBM) [18, 19].

Missing proofs and details are relegated to the Appendix due to space limitations.

## 2    Related works

**DP for neural networks and classical graph privacy analysis.** Providing rigorous privacy guarantees for ML methods is a problem of significant interest [20–23], where DP gradient descent algorithms and their variants were studied in [14]. On the other hand, classical graph analysis with DP guarantees has been extensively studied previously. For example, problems of releasing graphs or their statistics with DP guarantees were examined in [24] and [25]. In a different setting, [26] investigated the problem of estimating degree distributions with DP guarantees, while [27] studied DP PageRank algorithms. The interested reader is referred to the survey [28] for a more comprehensive list of references.

**DP-GNNs.** Several attempts were made to establish formal GNN privacy guarantees via DP. [29] propose a strategy achieving edge DP by privatizing the graph structure (i.e., perturbing the topology) before feeding it into GNNs. [15] describe a node DP approach for training general GNNs via extensions of DP-SGD, while [30] suggests to combine DP PageRank with DP-SGD GNN training. These approaches can only guarantee DP model weights and fail to provide DP model predictions. [31] study GNNs for graph classification instead of node classification. [32] propose node-level private GNNs via the Private Aggregation of Teacher Ensembles framework [33], which is a different setting than ours. [17] are the first to point out the importance of ensuring DP of GNN predictions, and to propose the GAP model for this purpose. Still, their work did not include a formal GDP analysis nor a study of the benefits of decoupled graph convolution designs.

## 3    Preliminaries

**Notation.** We reserve bold-font capital letters (e.g., $\mathbf{S}$) for matrices and bold-font lowercase letters (e.g., $\mathbf{s}$) for vectors. We use $\mathbf{S}_i$ to denote the $i^{th}$ row of $\mathbf{S}$, $\mathbf{S}_{\backslash r}$ to denote the submatrix of $\mathbf{S}$ excluding its $r^{th}$ row and $\mathbf{S}_{ij}$ to denote the entry of $\mathbf{S}$ in the $i^{th}$ row and $j^{th}$ column. Furthermore, $\mathcal{G} = (\mathcal{V}, \mathcal{E})$ stands for a directed graph with node set $\mathcal{V} = [n]$ of size $n$ and edge set $\mathcal{E}$. For simplicity, we assume that the graphs do not have self-loops. Standardly, we use $\mathbf{A}$ to denote the corresponding adjacency matrix. Without loss of generality, we also assume a labeling such that the first $m$ nodes are labeled ($[m]$) and the remaining $[n] \setminus [m]$ nodes are subjects of our prediction. The feature matrix is denoted by $\mathbf{X} \in \mathbb{R}^{n \times F}$, where $F$ stands for the feature vector dimension. For a $C$-class node classification problem, the training labels are summarized in $\mathbf{Y} \in \{0, 1\}^{m \times C}$, where each row of $\mathbf{Y}$ is a one-hot vector. We reserve $\| \cdot \|$ for the $\ell_2$ norm and $\| \cdot \|_F$ for the Frobenius norm. We use $\|$ for concatenation. Throughout the paper, we let $\mathcal{M}(v; \mathcal{D})$ stand for the (graph) learning algorithm that leverages $\mathcal{D}$ to generate label predictions for a node $v$ in $[n] \setminus [m]$. We assume the graph model outputs both its learned weights $\mathbf{W}$ and its label prediction $\widehat{\mathbf{Y}}_v$ (i.e., $(\widehat{\mathbf{Y}}_v, \mathbf{W}) = \mathcal{M}(v; \mathcal{D})$).

**Node classification.** We focus on the transductive node classification problem. The training data is of the form $\mathcal{D} = (\mathbf{X}, \mathbf{Y}, \mathbf{A})$, and this information is reused at the inference stage. Due to data reusing, it is important to specify which node $v$ is subject to prediction in the graph learning mechanism $\mathcal{M}(v; \mathcal{D})$. Requiring that all of $\mathcal{D}$ is private inevitably leads to noninformative predictions. It is therefore reasonable to only protect the information of $\mathcal{D}$ pertaining to the node that is not subject to prediction. We refer interested readers to Appendix I for discussion on how our analysis generalizes to inductive settings.

## 3.1 Differential Privacy (DP)

We start with a formal definition of $(\varepsilon, \delta)$-differential privacy (DP) [13].

**Definition 3.1** (Differential Privacy). *For $\varepsilon, \delta \geq 0$, a randomized algorithm $\mathcal{A}$ satisfies a $(\varepsilon, \delta)$-DP condition if for all adjacent datasets $\mathcal{D}, \mathcal{D}'$ that differ in one record, and all $\mathcal{S}$ in the range of $\mathcal{A}$,*

$$\Pr\left(\mathcal{A}(\mathcal{D}) \in \mathcal{S}\right) \leq e^{\varepsilon} \Pr\left(\mathcal{A}(\mathcal{D}') \in \mathcal{S}\right) + \delta.$$

Note that the standard DP definition does not depend on the choice of nodes to be predicted. Yet, such a dependence is critical for the graph learning setting and constitutes the key difference between Definition 3.1 and our definition of GDP in Section 4. We also make use of Rényi DP in order to facilitate privacy accounting. A given $(\alpha, \gamma)$-Rényi DP guarantee can be convert to a $(\varepsilon, \delta)$-DP guarantee via the conversion lemma [34–36] (see also Appendix F).

**Definition 3.2** (Rényi Differential Privacy). *Consider a randomized algorithm $\mathcal{A}$ that takes $\mathcal{D}$ as its input. The algorithm $\mathcal{A}$ is said to be $(\alpha, \gamma)$-Rényi DP if for every pair of adjacent datasets $\mathcal{D}$ and $\mathcal{D}'$, one has $D_\alpha(\mathcal{A}(\mathcal{D})||\mathcal{A}(\mathcal{D}')) \leq \gamma$, where $D_\alpha(\cdot||\cdot)$ denotes the Rényi divergence of order $\alpha$:*

$$D_\alpha(X||Y) = \frac{1}{\alpha - 1} \log\left(\mathbb{E}_{x \sim Q}\left[\left(\frac{P(x)}{Q(x)}\right)^\alpha\right]\right), \text{ with } X \sim P \text{ and } Y \sim Q.$$

# 4 Graph Differential Privacy

We provide next a formal definition of Graph Differential Privacy (GDP). Recall that in this case keeping the entire training data $\mathcal{D}$ private (as in standard DP definition, i.e., Definition 3.1) is problematic since it inevitably leads to noninformative model predictions. This follows since in such a setting the prediction $\widehat{\mathbf{Y}}_v$ fully depends on $\mathcal{D}$. This is unlike the case for standard classification where the test label predictions are formed via access to additional test data that is not subject to privacy constraints. Our key idea is to protect the privacy of all but the node $v$ being predicted, as users who query their own predictions should clearly have access to their own features and neighbors.

We start by describing our notion of $k$-neighbor-level adjacent graph datasets. Throughout the remainder of the paper, we use $r$ to denote the replaced node.

**Definition 4.1** ($k$-neighbor-level adjacency). *Two graph datasets $\mathcal{D}$ and $\mathcal{D}'$ are said to be $k$-neighbor-level adjacent, which is denoted by $\mathcal{D} \overset{N_k}{\sim} \mathcal{D}'$, if $\mathcal{D}'$ can be obtained by replacing: 1) the $r^{th}$ node feature $\mathbf{X}_r$ with $\mathbf{X}'_r \in \mathbb{R}^d$, 2) the $r^{th}$ node label $\mathbf{Y}_r$ with $\mathbf{Y}'_r \in \{0, 1\}^C$, for $r \in [m]$, and 3) replacing $k$ entries in $r^{th}$ row and column of $\mathbf{A}$ respectively excluding $\mathbf{A}_{rr}$.*

Our Definition 4.1 unifies previous graph dataset adjacency notions such as edge- and node-level adjacency. For example, if we drop 1), 2) (i.e., keep the node attributes unchanged), and set $k = 1$ for replacement in a row only, our Definition 4.1 recovers that of edge-level adjacency [29]. If $k = n$, we recover the definition of node-level adjacency [15, 17, 37, 38]. Edge-level adjacency may be too weak of a privacy concept since it does not protect the privacy of node attributes. At the same time, node-level adjacency may also be too restrictive since it allows arbitrary replacement of an *entire* node neighborhood. In practice, there are cases where node features and labels carry more sensitive information when compared to the graph topology. It is desirable to allow practitioners to decide the granularity of privacy for the graph structure $\mathbf{A}$ while maintaining the privacy of $\mathbf{X}$ and $\mathbf{Y}$. This motivates our new and extended $k$-neighbor-level adjacency definition. Note that the parameter $k$ serves as a new graph-specific privacy parameter similar to, but independent of, $\epsilon$ and $\delta$ in DP. Our $k$-neighbor definition can shed light on how a private graph learning design relies on the privacy of $\mathbf{A}$ controlled by the parameter $k$ and is discussed in more detail in Section 5. We also provide a

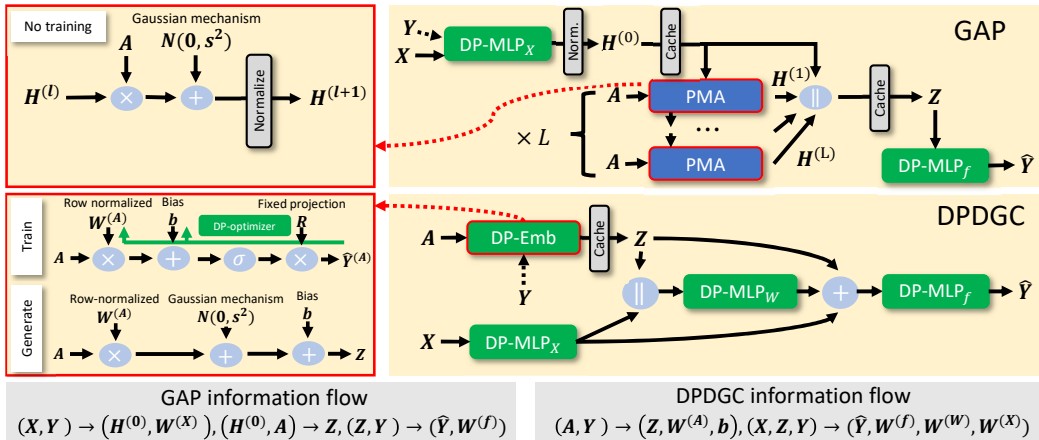

Figure 2: Illustration of the GAP and DPDGC (top) architectures and their corresponding information flow (bottom). Green modules indicate DP-MLPs trained with a DP-optimizer [14]. Blue modules are non-trainable modules. We use red frames to point to designs with DP guarantees (i.e., DP-Emb and PMA [17]). Trainable weights are denoted by $\mathbf{W}^{(A)}$ and $\mathbf{b}$ for the DP-Emb module. The black dashed arrow indicates modules that are pretrained separately and the outputs are cached.

detailed discussion on the privacy meaning of $k$ in Appendix J. In practice, the parameter $k$ also offers a unique trade-off between utility and graph structure privacy while preserving the same privacy guarantees of the node features $\mathbf{X}$ and labels $\mathbf{Y}$ (i.e., it is independent of the DP parameters $\epsilon, \delta$).

We now provide our formal definition of Graph Differential Privacy (GDP). Unless otherwise specified, we use the superscript $'$ to refer to entities with respect to the adjacent dataset $\mathcal{D}'$.

**Definition 4.2** (Graph Differential Privacy). *A graph model $\mathcal{M}$ is said to be edge (node, $k$-neighbor) $(\alpha, \gamma)$-GDP if for any $v \in [n] \setminus [m]$, for all $\mathcal{D} \overset{E}{\sim} \mathcal{D}'$ ($\mathcal{D} \overset{N}{\sim} \mathcal{D}'$, $\mathcal{D} \overset{N_k}{\sim} \mathcal{D}'$), such that $r \neq v$, one has: $D_\alpha(\mathcal{M}(v; \mathcal{D}) || \mathcal{M}(v; \mathcal{D}')) \leq \gamma$, where $r$ is the index of the replaced node in the dataset pair $\mathcal{D}, \mathcal{D}'$.*

The key difference between our definition and that of standard DP is that instead of requiring the Rényi divergence bound to hold for all possible $(\mathcal{D}, \mathcal{D}')$ pairs, we only require it to hold for pairs for which the replaced node $r$ does not require prediction (i.e., $r \neq v$). This is crucial since in this case, the Rényi divergence has to be bounded for *different sets of adjacent graph dataset pairs that depend on $v$*. At a high level, Definition 4.2 ensures that even if an adversary obtains the trained weights and the predictions of node $v$, it cannot infer information about the remaining nodes.

## 5 Graph learning methods with GDP guarantees

We first perform the GDP analysis for GAP [17], the state-of-the-art DP-GNN with standard graph convolution design in Section 5.1. In the same section, issues with standard graph convolutions under $k$-neighbor GDP settings are discussed as well. We then proceed to introduce Differentially Private Decoupled Graph Convolution (DPDGC), a model with the decoupled graph convolution design that resolves all issues with GDP guarantees. DPDGC is motivated by the LINKX model [39] which offers excellent performance on heterophilic graphs in a non-private setting. These models are depicted in Figure 2, with the pseudocodes available in Appendix L. All missing formal statements and proofs are relegated to Appendix B- E. We also defer the analysis of the simper edge GDP scenario to Appendix G and formal GDP guarantees to Appendix H.

### 5.1 GDP guarantees of GAP and issues of standard graph convolution

**GAP training and inference.** We first describe the training process of GAP depicted in Figure 2. GAP first pretrains the node feature encoder DP-MLP$_X$ separately from the DP-optimizer. The row-normalized node embedding $\mathbf{H}^{(0)}$ is generated and cached after the pretraining of DP-MLP$_X$. Then the privatized $L$ multi-hop results $\{\mathbf{H}^{(l)}\}_{l=0}^{L}$ are generated by applying the PMA module [17]. The intermediate node embedding $\mathbf{Z}$ constructed by the concatenation of $\{\mathbf{H}^{(l)}\}_{l=0}^{L}$ is then cached.

Finally, a node classifier DP-MLP$_f$ is trained with input $\mathbf{Z}$. At the inference stage, node predictions are obtained by the cached embedding $\mathbf{Z}$ with the trained DP-MLP$_f$.

**Node and $k$-neighbor GDP.** We assume that the out-degree (column-sum) of $\mathbf{A}$ is bounded from above by $D$. Note that prior works [15, 17] also require this assumption. To meet this constraint in practice, preprocessing of the form of graph subsampling is needed, which causes additional data distortion. The first step of proving tight GDP guarantees for GAP is to ensure the cached embedding $\mathbf{Z}$ to be DP, *except for the replaced node $r$*. We start with describing the PMA [17] module:

$$\text{Input: } \mathbf{H}^{(l)} \in \mathbb{R}^{n \times h}; \quad \text{Output: } \mathbf{H}^{(l+1)} = \text{row-normalization}(\mathbf{A}\mathbf{H}^{(l)} + \mathbf{N}), \tag{1}$$

where $\mathbf{N} \in \mathbb{R}^{n \times h}$ is a Gaussian noise matrix whose entries are i.i.d. zero mean Gaussian random variables with standard deviation $s$.

**Theorem 5.1.** *For any $\alpha > 1$ and $\mathcal{D} \overset{N}{\sim} \mathcal{D}'$ (or $\mathcal{D} \overset{N_k}{\sim} \mathcal{D}'$), assume the trained parameter $\mathbf{W}^{(X)}$ of DP-MLP$_X$ in GAP satisfies $D_\alpha(\mathbf{W}^{(X)}||\mathbf{W}^{(X)'}) \leq \gamma_1$ and that both $\mathbf{A}$, $\mathbf{A}'$ have out-degree bounded by $D$. Let the replaced node index be $r$ and let $\mathbf{Z}_{\backslash r}$ be the matrix $\mathbf{Z}$ with the $r^{th}$ row excluded. Then the embedding $\mathbf{Z}$ in GAP satisfies $D_\alpha(\mathbf{Z}_{\backslash r}||\mathbf{Z}'_{\backslash r}) \leq \gamma_1 + \frac{4DL\alpha}{2s^2}$.*

*Sketch of the proof:* We start by showing that for any $\mathcal{D} \overset{N}{\sim} \mathcal{D}'$, $D_\alpha(\mathbf{H}^{(1)}_{\backslash r}||\mathbf{H}^{(1)'}_{\backslash r}) \leq \gamma_1 + \frac{4D\alpha}{2s^2}$, which is done by examining the sensitivity of $[\mathbf{A}\mathbf{H}^{(0)}]_{\backslash r}$. For simplicity, we abbreviate $\mathbf{H}^{(0)}$ to $\mathbf{H}$. Note that $\left\|[\mathbf{A}\mathbf{H}]_{\backslash r} - [\mathbf{A}'\mathbf{H}']_{\backslash r}\right\|_F^2 = \sum_{i \in [n]\backslash\{r\}} \|[\mathbf{A}\mathbf{H}]_i - [\mathbf{A}'\mathbf{H}']_i\|^2$. We find that there are three cases of $i$ that contribute to a nonzero norm in the summation. Let $N(r)$ and $N'(r)$ denote the out-neighbor node sets of $r$ with respect to $\mathbf{A}$ and $\mathbf{A}'$, respectively (i.e., $N(r) = \{i : \mathbf{A}_{ir} = 1\}$). The three cases are: (1) $i \in N(r) \backslash N'(r)$, (2) $i \in N'(r) \backslash N(r)$, and (3) $i \in N(r) \cap N'(r)$. For case (1) and (2), $\|[\mathbf{A}\mathbf{H}]_i - [\mathbf{A}'\mathbf{H}']_i\| = 1$ due to $\mathbf{H}$ and $\mathbf{H}'$ being row-normalized. For case (3), we have $\|[\mathbf{A}\mathbf{H}]_i - [\mathbf{A}'\mathbf{H}']_i\| = \|\mathbf{H}_r - \mathbf{H}'_r\| \leq 2$. Since the out-degree is upper bounded by $D$, we know that $\max(|N(r)|, |N'(r)|) \leq D$. The worst case arises for $|N(r) \cap N'(r)| = D$ (i.e., shares common neighbors). Thus the sensitivity equals $2\sqrt{D}$ (i.e., $D$ of case (3)), which leads to the term $\frac{4D\alpha}{2s^2}$ in the divergence bound. By applying DP composition theorem [40] and the assumption on $\mathbf{W}^{(X)}$, we can prove that $D_\alpha(\mathbf{H}^{(1)}_{\backslash r}||\mathbf{H}^{(1)'}_{\backslash r}) \leq \gamma_1 + \frac{4D\alpha}{2s^2}$. For the $L$-hop result $\mathbf{Z}_{\backslash r}$, one can apply DP composition theorem as part of an induction. For the case of $k$-neighbor-level adjacency, the worst case scenario still arises for $|N(r) \cap N'(r)| = D$ for any $k \geq 0$. Thus, the same result holds for the case $k$-neighbor-level adjacency for all $k \geq 0$. Note that the above analysis/result is tight, with the worst case arising when $\mathbf{H}_r = -\mathbf{H}'_r$ and $\max(|N(r)|, |N'(r)|) = D$.

Regarding the assumption on $\mathbf{W}^{(X)}$, it can be met by invoking a standard DP-optimizer result [14], where $\gamma_1$ depends on the noise multiplier of the DP-optimizer. By using further the DP composition theorem and standard DP-optimizer results, we can conclude that the weights of the DP-MLP$_f$ module are also DP. At the inference stage, since our GDP definition requires that $r \neq v$ (i.e., nodes to be predicted are not subject to replacement in adjacent graph dataset pairs), one only needs to ensure that $\mathbf{Z}_{\backslash r}$ – instead of the entire $\mathbf{Z}$ – to be DP. This establishes the node and $k$-neighbor GDP guarantees for GAP (Corollary H.5).

Surprisingly, the resulting divergence bound does not depend on the parameter $k$ for the $k$-neighbor GDP setting. It implies that the privacy noise scale $s$ required by GAP is the same even when $k$ is much smaller than $D$, or even equal to zero. At first, this result seems counter-intuitive but can be understood as follows. By replacing $\mathbf{H}_r$ with an all-zeros row, it becomes impossible to get information about the $r^{th}$ row and column of $\mathbf{A}$ through $[\mathbf{A}\mathbf{H}]_{\backslash r}$. Therefore, DP-GNNs based on standard graph convolution designs such as GAP cannot exploit the intrinsic privacy-utility trade-offs induced by $k$-neighbor-level adjacency constraints. Furthermore, the resulting divergence bound grows linearly to the value of the maximum degree $D$. Consequently, one has to preprocess the graph so that the maximum degree is upper-bounded by a pre-defined value $D$, which inevitably causes graph information distortion.

Based on our analysis, we observe that the issue arises from the graph convolution operation $\mathbf{A}\mathbf{H}$, where both the graph structure (topology) $\mathbf{A}$ and transformed node feature $\mathbf{H}$ change simultaneously to $\mathbf{A}'$ and $\mathbf{H}'$ on $\mathcal{D}'$. As a result, rows corresponding to case (3) contribute 2 to the norm bound, which indicates greater privacy leakage (sensitivity). This motivates us to decouple the graph convolution

so that there are no products of $\mathbf{A}$ and $\mathbf{H}$ to work with, which motivates introducing our DPDGC model discussed in the next section.

**Remark 5.2.** *While the proof of Theorem 5.1 is mainly inspired by the proof in [17], it has several technical differences. First, the analysis in [17] asserts that the **entire** $\mathbf{Z}$ is DP (see Lemma 3 in [17]). In contrast, we only ensure that $\mathbf{Z}_{\backslash r}$ is DP. This is crucial as $\|[\mathbf{AH}]_r - [\mathbf{A'H'}]_r\| = 2D$ in the worst-case sensitivity analysis, which leads to $O(D^2)$ in the divergence bound. Also, while we leverage standard DP composition theorems [40] in the sketch of proof similar to [17] for the sake of simplicity, we argue in Appendix B that it is more appropriate to use our novel generalized adaptive composition theorem (Theorem B.1) for a rigorous GDP analysis.*

## 5.2 GDP guarantees of DPDGC and benefits of decoupled graph convolution

**DPDGC training and inference.** We first describe the training process of DPDGC depicted in Figure 2. We first pretrain the DP-Emb module separately and freeze its weights $(\mathbf{W}^{(A)}, \mathbf{b})$. Then we generate the intermediate embedding $\mathbf{Z}$ and cache it. Finally, we use $(\mathbf{X}, \mathbf{Z}, \mathbf{Y})$ to train the remaining modules in an end-to-end fashion. At the inference stage, the node prediction is obtained by $(\mathbf{X}, \mathbf{Z})$ and the trained weights. See the pseudocode in Appendix L for further details.

**Node $k$-neighbor GDP guarantees.** For DPDGC, we only need the bounded out-degree assumption for node GDP but not $k$-neighbor GDP. The key idea of the GDP guarantee proof is to ensure the cached embedding $\mathbf{Z}$ to be DP, *except for the replaced node $r$*. We start by introducing the DP-Emb module in DPDGC, which guarantees both the model weight $(\mathbf{W}^{(A)}, \mathbf{b})$ and $\mathbf{Z}_{\backslash r}$ to be DP:

$$\text{Training: } \widehat{\mathbf{Y}}^{(A)} = \sigma(\mathbf{AW}^{(A)} + \mathbf{b})\mathbf{R} \quad \text{Generate } \mathbf{Z}: \mathbf{Z} = \mathbf{AW}^{(A)} + \mathbf{b} + \mathbf{N}, \tag{2}$$

where $\mathbf{W}^{(A)} \in \mathbb{R}^{n \times h}$ and $\mathbf{b} \in \mathbb{R}^h$ are learnable weights. Here, $\mathbf{R} \in \mathbb{R}^{h \times C}$ is a random but fixed projection head of hidden dimension $h$, $\sigma(\cdot)$ is some nonlinear activation function, and $\mathbf{N} \in \mathbb{R}^{n \times h}$ is a Gaussian noise matrix whose entries are i.i.d. zero mean Gaussian random variables with standard deviation $s$. Importantly, we constraint $\mathbf{W}^{(A)}$ to be row-normalized, which is critical in proving $\mathbf{Z}_{\backslash r}$ DP. In what follows, we focus on privatizing $\mathbf{Z}_{\backslash r}$, and the GDP guarantee for the overall model then follows by applying DP composition theorem [40].

**Theorem 5.3.** *For any $\alpha > 1$ and $\mathcal{D} \overset{N}{\sim} \mathcal{D}'$, assume that $D_\alpha((\mathbf{W}^{(A)}, \mathbf{b})\|(\mathbf{W}^{(A)}, \mathbf{b})') \leq \gamma_1$ and both $\mathbf{A}$, $\mathbf{A}'$ have out-degree bounded by $D$. Let the replaced node index be $r$ and let $\mathbf{Z}_{\backslash r}$ be the matrix $\mathbf{Z}$ with the $r^{th}$ row excluded. Then the embedding $\mathbf{Z}$ in DPDGC satisfies $D_\alpha(\mathbf{Z}_{\backslash r}\|\mathbf{Z}'_{\backslash r}) \leq \gamma_1 + \frac{2D\alpha}{2s^2}$.*

**Theorem 5.4.** *For any $\mathcal{D} \overset{N_k}{\sim} \mathcal{D}'$, assume that $D_\alpha((\mathbf{W}^{(A)}, \mathbf{b})\|(\mathbf{W}^{(A)}, \mathbf{b})') \leq \gamma_1$ and both $\mathbf{A}$. Let the replaced node index be $r$ and let $\mathbf{Z}_{\backslash r}$ be the matrix $\mathbf{Z}$ with the $r^{th}$ row excluded. For any $\alpha > 1$, the embedding $\mathbf{Z}$ in DPDGC satisfies $D_\alpha(\mathbf{Z}_{\backslash r}\|\mathbf{Z}'_{\backslash r}) \leq \gamma_1 + \frac{k\alpha}{2s^2}$.*

*Sketch of the proof:* For any $\mathcal{D} \overset{N}{\sim} \mathcal{D}'$ ($\mathcal{D} \overset{N_k}{\sim} \mathcal{D}'$), we examine $\|\left[\mathbf{AW}^{(A)}\right]_i - \left[\mathbf{A'W}^{(A)}\right]_i\|$, $i \in [n] \backslash \{r\}$. The worst case row norm equals 1 for cases (1) and (2) (defined in the proof of Theorem 5.1) since $\mathbf{W}^{(A)}$ is row-normalized. The main difference to the proof in Section 5.1 is the case (3), where $\|\left[\mathbf{AW}^{(A)}\right]_i - \left[\mathbf{A'W}^{(A)}\right]_i\| = 0$, since $\mathbf{A}_i = \mathbf{A}'_i$ for $i \in N(r) \cap N'(r)$. As a result, the worst case arises for $|N(r) \cap N'(r)| = \emptyset$ and there are at most $2D$ ($k$) rows of cases (1) and (2). As a result, the sensitivity of $\mathbf{Z}_{\backslash r}$ is $\sqrt{2D}$ ($\sqrt{k}$) which leads to the term $\frac{2D\alpha}{2s^2}$ ($\frac{k\alpha}{2s^2}$) in the divergence bound. Once more, our sensitivity bound can be shown to be tight, with the worst case as described above.

The key improvement, when compared to GAP, arises from using $\mathbf{AW}^{(A)}$ instead of $\mathbf{AH}$. In the case of $\mathbf{W}^{(A)}$ being DP, the sensitivity analysis only requires one to consider the difference between $\mathbf{AW}^{(A)}$ and $\mathbf{A'W}^{(A)}$ of two adjacent graph datasets, $\mathcal{D}$ and $\mathcal{D}'$. In contrast, for GAP, the sensitivity analysis of $\mathbf{AH}$ requires taking *both* $\mathbf{A}$ and $\mathbf{H}$ into account since both can vary when the underlying dataset changes. Note that even if DP-MLP$_X$ is trained via a DP-optimizer, $\mathbf{H}$ is not DP since it is dependent on the weights of DP-MLP$_X$ and the node feature $\mathbf{X}$.

The assumption on $(\mathbf{W}^{(A)}, \mathbf{b})$ can be met by applying a standard DP-optimizer with a group size [41] of $D + 1$ ($k + 1$), where $\gamma_1$ depends on the noise multiplier of the DP-optimizer. This follows from the fact that the out-degree is bounded by $D$ for node GDP or from the definition of $k$-neighbor-level adjacency, since replacing one node can affect at most $D$ ($k$) neighbors. By further applying the DP

Table 1: Dataset statistics. Datasets are sorted by homophily.

|  | Squirrel | Chameleon | Facebook | Pubmed | Computers | Cora | Photo |
|---|---|---|---|---|---|---|---|
| nodes | 5,201 | 2,277 | 26,406 | 19,717 | 13,471 | 2,708 | 7,535 |
| edges | 216,933 | 36,051 | 2,117,924 | 88,648 | 491,722 | 10,556 | 238,162 |
| features | 2,089 | 2,325 | 501 | 500 | 767 | 1,433 | 745 |
| classes | 5 | 5 | 6 | 3 | 10 | 7 | 8 |
| edge density | 0.0160 | 0.0139 | 0.0061 | 0.0005 | 0.0054 | 0.0029 | 0.0084 |
| homophily | 0.0254 | 0.0620 | 0.3687 | 0.6641 | 0.7002 | 0.7657 | 0.7722 |

composition theorem and the standard DP-optimizer result, we arrive at the node ($k$-neighbor) GDP guarantees for DPDGC (see Corollary H.2, H.3).

Compared to GAP, DPDGC requires significantly lower DP noise whenever $k < D$. It can thus ensure a graph topology privacy-utility trade-off that is not possible with GAP. Furthermore, the divergence bound for DPDGC is independent of the maximum degree within the $k$-neighbor-level adjacency setting. Hence, DPDGC does not require preprocessing the graph, which alleviates the issue of added graph distortion.

## 6 Experiments

We test graph learning models that can achieve GDP guarantees under various settings, including nonprivate, edge-level, $k$-neighbor-level ($N_k$) for $k \in \{1, 5, 25\}$, and node-level. Note that all node GDP methods require the bounded out-degree constraint. We follow [15, 17] to subsample the graph so as to satisfy this constraint.

**Methods.** In addition to DPDGC and GAP introduced in Section 5, we also test (DP-)MLP and several DP-GNN baselines that can achieve GDP guarantees, including RandEdge+SAGE [29] and DP-SAGE [15] for edge and node GDP, respectively. The RandEdge+SAGE approach privatizes the adjacency matrix directly via randomized response [29] and feeds it to GraphSAGE [3], a standard GNN backbone. The DP-SAGE approach trains the GraphSAGE model with the strategy proposed in [15] so that the weights are DP. To further ensure that the predictions are DP as well, we follow the strategy of [17] to add perturbations directly at the output layer during inference. For each GDP setting, we specify $\epsilon$ to indicate that all methods satisfy GDP with privacy budget $(\epsilon, \delta)$ according to Lemma F.1. Note that $\delta$ is set to be smaller than either $\frac{1}{\#\text{edges}}$ or $\frac{1}{\#\text{nodes}}$, depending on the GDP setting. Addition experimental details are deferred to Appendix K.

**Datasets.** We test 7 benchmark datasets available from either Pytorch Geometric library [42] or prior works. These datasets include the social network **Facebook** [43], citation networks **Cora** and **Pubmed** [44, 45], Amazon co-purchase networks **Photo** and **Computers** [46], and Wikipedia networks **Squirrel** and **Chameleon** [47]. The edge density equals $2|\mathcal{E}|/(n \times (n-1))$, while the formal definition of the homophily measure is relegated to Appendix K.

**Results.** We examine the performance of each model under different GDP settings, including the nonprivate case. The results are summarized in Table 2. For the edge GDP setting, we observe that DPDGC achieves the best performance on heterophilic datasets, while on par with GAP on most homophilic datasets. Surprisingly, for the node GDP setting, we find that DP-MLP achieves the best performance on four datasets. This indicates that for some datasets, the benefits of graph information cannot compensate for the utility loss induced by privacy noise that protects the graph information (for all tested private graph learning methods). As a result, achieving node GDP effectively is highly challenging and highlights the importance of investigating the $k$-neighbor GDP setting. For the $k$-neighbor GDP setting, we can see that indeed DPDGC has a much better utility for $k = 1$ and the performance gradually decreases as $k$ grows. The required privacy noise scale is the same for GAP and DP-MLP for any $k$: this phenomenon is discussed in more depth in Section 5. DPDGC starts to outperform DP-MLP on Photo and Computers for sufficiently small $k$, which demonstrates the unique utility-topology privacy trade-offs of our method that cannot be achieved via models with standard graph convolution design such as GAP. However, DPDGC still underperforms compared to DP-MLP on Cora and Pubmed.

**Experiments on synthetic contextual stochastic block models (cSBMs).** In order to test our conjecture that DP-MLP can only outperform DPDGC when the graph information is "too weak," we conduct an experiment involving cSBMs [18], following a setup similar to that reported in [19].

Table 2: Test accuracy (%) with 95% confidence interval. A bold font indicates the best performance one can achieve under the same GDP guarantee. Underlined entries indicate a result within the confidence interval when compared to the best possible. Note that for the $k$-neighbor ($N_k$) GDP setting, the results of GAP and DP-MLP are identical to those of node GDP.

| | | Squirrel | Chameleon | Facebook | Pubmed | Computers | Cora | Photo |
|---|---|---|---|---|---|---|---|---|
| none private | DPDGC | **79.92 ± 0.26** | **79.24 ± 0.54** | **86.06 ± 0.24** | 88.34 ± 0.46 | **92.27 ± 0.15** | 82.44 ± 0.83 | 94.98 ± 0.29 |
| | GAP | 36.86 ± 1.35 | 50.35 ± 1.37 | 79.52 ± 0.24 | **89.75 ± 0.12** | 91.05 ± 0.16 | **86.53 ± 0.46** | **95.13 ± 0.16** |
| | SAGE | 35.47 ± 0.58 | 41.61 ± 0.86 | 84.62 ± 0.13 | 88.17 ± 0.98 | 91.76 ± 0.23 | 84.19 ± 0.76 | 94.05 ± 0.38 |
| | MLP | 34.14 ± 0.77 | 46.78 ± 1.39 | 51.16 ± 0.16 | 87.25 ± 0.19 | 85.27 ± 0.28 | 76.48 ± 0.91 | 91.35 ± 0.22 |
| edge ε = 1 | DPDGC | **38.18 ± 1.48** | **53.83 ± 1.11** | 62.04 ± 0.33 | **88.59 ± 0.16** | **87.74 ± 0.26** | **77.71 ± 0.95** | 92.59 ± 0.41 |
| | GAP | 35.15 ± 0.47 | 49.47 ± 0.88 | **69.75 ± 0.44** | 87.79 ± 0.22 | **87.74 ± 0.20** | 76.95 ± 0.90 | **92.94 ± 0.36** |
| | RandEdge+SAGE | 19.79 ± 0.69 | 21.70 ± 1.23 | 25.27 ± 2.00 | 87.88 ± 0.18 | 48.44 ± 1.48 | 59.95 ± 1.98 | 46.42 ± 0.55 |
| | DP-MLP | 34.14 ± 0.77 | 46.78 ± 1.39 | 51.16 ± 0.16 | 87.25 ± 0.19 | 85.27 ± 0.28 | 76.48 ± 0.91 | 91.35 ± 0.22 |
| $N_1 = 16$ ε = 16 | DPDGC | **42.71 ± 1.43** | **48.63 ± 1.78** | **80.94 ± 0.27** | 84.33 ± 0.40 | **83.49 ± 0.29** | 59.98 ± 0.81 | **88.38 ± 0.44** |
| | GAP | 33.82 ± 0.60 | 38.68 ± 0.59 | 51.57 ± 0.28 | 85.28 ± 0.14 | 77.50 ± 0.20 | 54.36 ± 1.14 | 81.27 ± 0.31 |
| | DP-MLP | 34.46 ± 1.09 | 38.19 ± 1.97 | 50.12 ± 0.22 | **85.72 ± 0.11** | 80.01 ± 0.37 | **64.29 ± 0.80** | 85.61 ± 0.42 |
| $N_5 = 16$ ε = 16 | DPDGC | **41.00 ± 1.19** | **47.22 ± 1.90** | **76.84 ± 0.36** | 84.31 ± 0.46 | **80.60 ± 0.44** | 59.56 ± 0.97 | **87.02 ± 0.49** |
| | GAP | 33.82 ± 0.60 | 38.68 ± 0.59 | 51.57 ± 0.28 | 85.28 ± 0.14 | 77.50 ± 0.20 | 54.36 ± 1.14 | 81.27 ± 0.31 |
| | DP-MLP | 34.46 ± 1.09 | 38.19 ± 1.97 | 50.12 ± 0.22 | **85.72 ± 0.11** | 80.01 ± 0.37 | **64.29 ± 0.80** | 85.61 ± 0.42 |
| $N_{25} = 16$ ε = 16 | DPDGC | **40.51 ± 0.85** | **46.32 ± 1.87** | **68.66 ± 0.32** | 84.27 ± 0.38 | 78.25 ± 0.31 | 59.26 ± 0.87 | 84.82 ± 0.54 |
| | GAP | 33.82 ± 0.60 | 38.68 ± 0.59 | 51.57 ± 0.28 | 85.28 ± 0.14 | 77.50 ± 0.20 | 54.36 ± 1.14 | 81.27 ± 0.31 |
| | DP-MLP | 34.46 ± 1.09 | 38.19 ± 1.97 | 50.12 ± 0.22 | **85.72 ± 0.11** | **80.01 ± 0.37** | **64.29 ± 0.80** | 85.61 ± 0.42 |
| node ε = 16 | DPDGC | **36.17 ± 0.62** | **46.43 ± 1.21** | **56.65 ± 0.64** | 84.55 ± 0.32 | 76.62 ± 0.47 | 58.97 ± 1.05 | 82.15 ± 0.54 |
| | GAP | 33.82 ± 0.60 | 38.68 ± 0.59 | 51.57 ± 0.28 | 85.28 ± 0.14 | 77.50 ± 0.20 | 54.36 ± 1.14 | 81.27 ± 0.31 |
| | DP-SAGE | 19.81 ± 0.99 | 20.96 ± 1.27 | 32.15 ± 0.78 | 39.68 ± 0.70 | 39.13 ± 0.52 | 15.86 ± 1.55 | 31.41 ± 0.93 |
| | DP-MLP | 34.46 ± 1.09 | 38.19 ± 1.97 | 50.12 ± 0.22 | **85.72 ± 0.11** | **80.01 ± 0.37** | **64.29 ± 0.80** | 85.61 ± 0.42 |

Figure 3: (a): Trade-off between utility and the parameter $\epsilon$ for edge-GDP on the Facebook dataset. (b): Trade-off between utility and the parameter $\epsilon$ for node and $k$-neighbor ($N_k$) GDP on the Facebook dataset. (c): Trade-off between utility and the parameter $k$ for $k$-neighbor GDP. (d): The cSBM experiment, which shows that when the graph information is strong enough (i.e., $|\phi|$ is large enough), DPDGC can indeed outperform DP-MLP.

The parameter $\phi \in [-1, 1]$ indicates the "strength of information" in the graph topology ($|\phi| \to 1$) and the node features ($|\phi| \to 0$). The sign of $\phi$ indicates whether the graph topology is homophilic ($+$) or heterophilic ($-$). See Appendix K for full details about the experiment setting. The result is depicted in Figure 3 (d). When $|\phi|$ is small, DP-MLP outperforms DPDGC. In contrast, when $|\phi|$ is sufficiently large (i.e., the information in the graph topology is strong enough), DPDGC outperforms DP-MLP. This experiment suggests that one should not leverage DP-GNNs when the graph topology information is too weak, as the cost of privatizing the graph topology information is too high.

**Utility-privacy trade-off.** Lastly, we examine the utility-privacy trade-off for different GDP settings in greater detail. In Figure 3 (left) we show that DPDGC starts to outperform GAP for larger privacy budgets – $\epsilon \geq 4$ – in the edge GDP setting. This figure also shows the advantage of GAP when the privacy budget $\epsilon$ is small for edge GDP. Whether a more advanced decoupled graph convolution design can outperform GAP in this regime is left for future studies. Figure 3 (middle) reports the trade-offs between utility and the privacy budget $\epsilon$ of all methods under different GDP settings (node, $N_1$, $N_5$, $N_{25}$) for the Facebook dataset. We report the trade-off between utility and $k$ under $k$-neighbor GDP with $\epsilon = 16$ for the Computer dataset in Figure 3 (right). Adopting the $k$-neighbor GDP definition allows us to control the privacy of the graph structure while maintaining the same privacy guarantee on node features and labels. This is in contrast with changing the parameters $(\epsilon, \delta)$ directly as they control the privacy of the *entire dataset*. DPDGC along with the $k$-neighbor GDP definitions hence allows a more fine-grained privacy control of the graph topology as needed in different practical applications.

# 7    Conclusions

We performed an analysis of a novel notion of Graph Differential Privacy (GDP), specifically tailored to graph learning settings. Our analysis established theoretical privacy guarantees for both model weights and predictions. In addition, to offer multigranular protection for the graph topology, we introduced the concept of $k$-neighbor-level adjacency, which is a relaxation of standard node-level adjacency. This allows for controlling the strength of privacy protection for node neighborhood information via a parameter $k$. The supporting GDP approach ensured a flexible trade-off between utility and topology privacy for graph learning methods. The GDP analysis also revealed that standard graph convolution designs failed to offer this trade-off. To provably mitigate the problem associated with standard convolutions, we introduced Differentially Private Decoupled Graph Convolution (DPDGC), a model that comes with GDP guarantees. Extensive experiments conducted on seven node classification benchmark datasets and synthetic cSBM datasets demonstrated the superior privacy-utility trade-offs offered by DPDGC when compared to existing DP-GNNs that rely on standard graph convolution designs.

## Acknowledgments and Disclosure of Funding

EC, CP and OM were funded by NSF grants CCF-1816913 and CCF-1956384. PL was supported by JPMC AI Research award. WC and AÖ were supported by NSF grant CCF-2213223. The authors would like to thank Sina Sajadmanesh for answering questions regarding the GAP method. The authors would like to thank the anonymous reviewers and area chair for their feedback and effort, which helped to significantly improve the manuscript.

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

## A  Limitations and Broader Impacts

**Limitations.** While our experimental results demonstrate several benefits of using DPDGC designs over standard convolutions, we do not believe that the current DPDGC model is the ultimate solution for GDP-aware graph learning methods. To support this claim, we note that the nonprivate state-of-the-art performance for learning on large-scale homophilic graphs is achieved by standard graph convolution models [48, 49]. Our experiments for edge GDP settings also reveal that standard graph convolution designs (such as GAP) appear to work well when the parameter $\epsilon$ is small. We therefore conjecture that further improvements for decoupled graph convolution designs are possible.

**Broader Impacts.** We are not aware of any negative social impacts of our work. In fact, our work establishes a formal DP framework for graph learning termed GDP. This is beneficial to many applications that require rigorous protection of the privacy of users when their sensitive information is stored as a graph dataset to be leveraged by graph learning methods. Note that DP is the gold standard for quantifying the privacy of user data and it has found widespread applications. Given the popularity of graph learning methods and the need for user privacy protection, we believe that our work will actually have a positive social impact.

## B  Generalized adaptive composition theorem

In order to ensure that a graph learning method satisfies GDP, we need to follow a commonly employed approach that involves the following steps: (1) introducing appropriate perturbations to the learned graph representation (e.g., $\mathbf{H}^{(0)}$ for GAP or $\mathbf{Z}$ for DP-DGC as shown in Figure 2), and (2) training the downstream DP-MLP using DPSGD. Subsequently, employing composition theorems allows for establishing an overall privacy budget. However, as we argued in Section 4, it is important to establish "partial DP" guarantees for intermediate node embeddings $\mathbf{H}^{(k)}$ or $\mathbf{Z}$ (i.e., ensure that only $\mathbf{Z}_{\backslash r}$ is DP and exclude $\mathbf{Z}_r$ from latter mechanisms, as in Theorem 5.3). When feeding the entire $\mathbf{Z}$ to the subsequent mechanism $\mathcal{B}(\mathbf{Z}_{\backslash r}, \mathbf{Z}_r)$, part of the input may not be DP and may exhibit correlations with other (processed) data components. This renders the standard composition theorem for DP [40, 41, 50] inapplicable. To address this issue, we developed a novel generalized composition theorem that can handle potential dependencies.

**Theorem B.1.** *Let $\mathcal{A} : \mathcal{D} \to (\mathcal{A}_1(\mathcal{D}), \mathcal{A}_2(\mathcal{D})) \in \mathcal{Z}_1 \times \mathcal{Z}_2$ satisfy the $(\alpha, \varepsilon_1)$-RDP constraint in its first output argument, i.e., $D_\alpha (\mathcal{A}_1(\mathcal{D}) \| \mathcal{A}_1(\mathcal{D}')) \le \varepsilon_1$. Let $\mathcal{B} : \mathcal{Z}_1 \times \mathcal{Z}_2 \to \mathcal{W}$ satisfy the $(\alpha, \varepsilon_2)$-RDP constraint in its second input argument, i.e., $\max_{\mathbf{z}_1, \mathbf{z}_2, \mathbf{z}_2'} D_\alpha (\mathcal{B}(\mathbf{z}_1, \mathbf{z}_2) \| \mathcal{B}(\mathbf{z}_1, \mathbf{z}_2')) \le \varepsilon_2$. Let the random noise used in $\mathcal{A}$ and $\mathcal{B}$ be independent. Then $(\mathcal{A}_1(\mathcal{D}), \mathcal{B}(\mathcal{A}(\mathcal{D})))$ jointly meets the $(\alpha, \varepsilon_1 + \varepsilon_2)$-RDP guarantee.*

*Proof.* For ease of explanation, we assume that the density (and the conditional density) of $\mathcal{A}$ and $\mathcal{B}$ with respect to the Lebesgue measure exist. Let

- $(\mathcal{A}_1(\mathcal{D}), \mathcal{A}_2(\mathcal{D})) := (\mathbf{Z}_1, \mathbf{Z}_2)$ and $(\mathcal{A}_1(\mathcal{D}'), \mathcal{A}_2(\mathcal{D}')) := (\mathbf{Z}_1', \mathbf{Z}_2')$;

- $\mathcal{B}(\mathcal{A}(\mathcal{D})) := \mathbf{W}$ and $\mathcal{B}(\mathcal{A}(\mathcal{D}')) := \mathbf{W}'$.

In addition, let $f_{\mathbf{W}}(\mathbf{w})$, $f_{\mathbf{Z}_1}(\mathbf{z}_1)$, and $f_{\mathbf{Z}_2}(\mathbf{z}_2)$ be the density of $\mathbf{W}, \mathbf{Z}_1$ and $\mathbf{Z}_2$, respectively. Observe that

$$D_\alpha\left((\mathbf{Z}_1, \mathbf{W}(\mathbf{Z}_1, \mathbf{Z}_2)) \| (\mathbf{Z}_1', \mathbf{W}(\mathbf{Z}_1', \mathbf{Z}_2'))\right)$$

$$= \frac{1}{\alpha-1} \log \mathbb{E}_{\mathbf{w}, \mathbf{z}_1}\left[\left(\frac{\int_{\mathbf{z}_2} f_{\mathbf{W}|\mathbf{Z}_1=\mathbf{z}_1, \mathbf{Z}_2=\mathbf{z}_2}(\mathbf{W}(\mathbf{z}_1, \mathbf{z}_2)=\mathbf{w}) \cdot f_{\mathbf{Z}_2|\mathbf{Z}_1=\mathbf{z}_1}(\mathbf{z}_2) \cdot f_{\mathbf{Z}_1}(\mathbf{z}_1) d\mathbf{z}_2}{\int_{\mathbf{z}_2} f_{\mathbf{W}|\mathbf{Z}_1'=\mathbf{z}_1, \mathbf{Z}_2'=\mathbf{z}_2'}(\mathbf{W}(\mathbf{z}_1, \mathbf{z}_2)=\mathbf{w}) \cdot f_{\mathbf{Z}_2'|\mathbf{Z}_1'=\mathbf{z}_1}(\mathbf{z}_2) \cdot f_{\mathbf{Z}_1'}(\mathbf{z}_1) d\mathbf{z}_2'}\right)^\alpha\right]$$

$$\overset{(a)}{\leq} \frac{1}{\alpha-1} \log \mathbb{E}_{\mathbf{z}_1}\left[\mathbb{E}_{\mathbf{w}}\left[\left(\frac{\int_{\mathbf{z}_2} f_{\mathbf{W}|\mathbf{Z}_1=\mathbf{z}_1, \mathbf{Z}_2=\mathbf{z}_2}(\mathbf{W}(\mathbf{z}_1, \mathbf{z}_2)=\mathbf{w}) \cdot f_{\mathbf{Z}_2|\mathbf{Z}_1=\mathbf{z}_1}(\mathbf{z}_2) d\mathbf{z}_2}{\int_{\mathbf{z}_2} f_{\mathbf{W}|\mathbf{Z}_1'=\mathbf{z}_1, \mathbf{Z}_2'=\mathbf{z}_2'}(\mathbf{W}(\mathbf{z}_1, \mathbf{z}_2)=\mathbf{w}) \cdot f_{\mathbf{Z}_2'|\mathbf{Z}_1'=\mathbf{z}_1}(\mathbf{z}_2) d\mathbf{z}_2'}\right)^\alpha \Bigg| \mathbf{z}_1\right]\right.$$
$$\left. \cdot \left(\frac{f_{\mathbf{Z}_1}(\mathbf{z}_1)}{f_{\mathbf{Z}_1'}(\mathbf{z}_1)}\right)^\alpha\right]$$

$$\overset{(b)}{\leq} \frac{1}{\alpha-1} \log \mathbb{E}_{\mathbf{z}_1}\left[\max_{\mathbf{z}_2, \mathbf{z}_2'}\left(\mathbb{E}_{\mathbf{w}}\left[\left(\frac{f_{\mathbf{W}|\mathbf{Z}_1=\mathbf{z}_1, \mathbf{Z}_2=\mathbf{z}_2}(\mathbf{W}(\mathbf{z}_1, \mathbf{z}_2)=\mathbf{w})}{f_{\mathbf{W}|\mathbf{Z}_1'=\mathbf{z}_1, \mathbf{Z}_2'=\mathbf{z}_2'}(\mathbf{W}(\mathbf{z}_1, \mathbf{z}_2)=\mathbf{w})}\right)^\alpha \Bigg| \mathbf{z}_1\right]\right) \cdot \left(\frac{f_{\mathbf{Z}_1}(\mathbf{z}_1)}{f_{\mathbf{Z}_1'}(\mathbf{z}_1)}\right)^\alpha\right]$$

$$\overset{(c)}{\leq} \frac{1}{\alpha-1} \log \mathbb{E}_{\mathbf{z}_1}\left[e^{\varepsilon_1 \cdot (\alpha-1)}\left(\frac{f_{\mathbf{Z}_1}(\mathbf{z}_1)}{f_{\mathbf{Z}_1'}(\mathbf{z}_1)}\right)^\alpha\right]$$

$$\overset{(d)}{\leq} \varepsilon_1 + \varepsilon_2,$$

where (a) holds by the tower rule of conditional expectations; (b) holds due to the joint quasi-convexity of the Rényi divergence [51]; (c) holds since $\mathcal{B}$ is $\varepsilon_2$ Rényi DP; and (d) holds because $\mathcal{A}_1$ is $\varepsilon_2$ Rényi DP. $\qquad\square$

**Remark B.2.** *Note that if $\mathcal{A}_2(\mathcal{D})$ is independent of $\mathcal{A}_1(\mathcal{D})$, then the above composition theorem reduces to the standard sequential composition [40], which follows directly from the chain rule for the Rényi divergence. However, in the analysis of GDP, we need a stronger result capable of handling the dependence between $\mathcal{A}_1(\mathcal{D})$ and $\mathcal{A}_2(\mathcal{D})$.*

In the privacy analysis of our proposed DPDGC, we set $\mathcal{A}_1(\mathcal{D})$ and $\mathcal{A}_2(\mathcal{D})$ in Theorem B.1 to $\mathbf{Z}_{\backslash r}$ and $\mathbf{Z}_r$, respectively, and let $\mathcal{B}(\mathcal{D})$ be the weights of the DP-MLP$_W$ trained via DP-SGD (see Appendix H). Nevertheless, we note that since the Gaussian noise samples added to $\mathbf{Z}_{\backslash r}$ and $\mathbf{Z}_r$ are indeed independent in DPDGC, by grouping $(\mathcal{A}_2(\mathcal{D}), \mathcal{B}(\mathcal{D}))$, one can directly apply the classical composition theorem [40] to $\mathcal{A}_1(\mathcal{D})$ and $(\mathcal{A}_2(\mathcal{D}), \mathcal{B}(\mathcal{D}))$ and arrive at the same conclusion. Similarly, in the one-hop GAP illustrated in Figure 2, we applied the same privacy analysis, with $\mathcal{A}_1(\mathcal{D})$, $\mathcal{A}_2(\mathcal{D})$, and $\mathcal{B}(\mathcal{D})$ set to $\mathbf{H}_{\backslash r}^{(0)}$, $\mathbf{H}_r^{(0)}$, and the weights of DP-MLP$_W$, respectively, which allowed us to obtain the GDP guarantee.

However, it is worth noting that for a $L$-hop GAP (i.e., when aggregation is performed $L$ times), the DP noise introduced in $\mathbf{H}_{\backslash r}^{(L)}$ and that introduced in $\mathbf{H}_r^{(L)}$ are no longer independent due to the aggregation step, and hence the standard composition theorem may not be applicable. In this case, our generalized composition theorem rigorously leads to the desired GDP guarantees.

In summary, our generalized composition theorem (Theorem B.1) provides a "cleaner argument" for the case that merely part of intermediate node embedding $\mathbf{Z}$ and $\mathbf{H}^{(k)}$ are DP (i.e., $\mathbf{Z}_{\backslash r}$ and $\mathbf{H}_{\backslash r}^{(k)}$ are DP but not the entire $\mathbf{Z}$ and $\mathbf{H}^{(k)}$). Furthermore, our Theorem B.1 also allows for handling more complicated cases (i.e., when the noise in $\mathcal{A}_1(\mathcal{D})$ and the one in $\mathcal{A}_2(\mathcal{D})$ are not independent).

## C  Proof of Theorem 5.3

**Theorem.** *For any $\alpha > 1$ and $\mathcal{D} \overset{N}{\sim} \mathcal{D}'$, assume that $D_\alpha((\mathbf{W}^{(A)}, \mathbf{b}) \| (\mathbf{W}^{(A)}, \mathbf{b})') \leq \gamma_1$ and that both $\mathbf{A}, \mathbf{A}'$ has bounded out-degree $D$. Let the replaced node index be $r$ and $\mathbf{Z}_{\backslash r}$ be the matrix $\mathbf{Z}$ excluding its $r^{th}$ row. Then the embedding $\mathbf{Z}$ in DPDGC satisfies $D_\alpha(\mathbf{Z}_{\backslash r} \| \mathbf{Z}_{\backslash r}') \leq \gamma_1 + \frac{2D\alpha}{2s^2}$.*

*Proof.* We start by examining $\|[\mathbf{A}'\mathbf{W}^{(A)}]_{\backslash r} - [\mathbf{A}\mathbf{W}^{(A)}]_{\backslash r}\|_F$. Note that

$$\|[\mathbf{A}'\mathbf{W}^{(A)}]_{\backslash r} - [\mathbf{A}\mathbf{W}^{(A)}]_{\backslash r}\|_F^2 = \sum_{i \in [n] \backslash \{r\}} \|(\mathbf{A}'_i - \mathbf{A}_i)\mathbf{W}^{(A)}\|^2,$$

which is the sum of row-norms of all nodes except for node $r$. Let us denote the out-neighborhood of node $r$ with respect to $\mathbf{A}$ as $N(r) = \{i : \mathbf{A}_{ir} = 1\}$. Similarly, we also have $N'(r) = \{i : \mathbf{A}'_{ir} = 1\}$. There are four possible cases for $i \in [n] \backslash \{r\}$: (1) $i \in N(r) \backslash N'(r)$, (2) $i \in N'(r) \backslash N(r)$, (3) $i \in N(r) \cap N'(r)$, and (4) $i \notin N(r) \cup N'(r)$. Clearly, for $i$ covered by cases (3) and (4), the corresponding row norm is 0 since $\mathbf{A}'_i = \mathbf{A}_i$ for $i \in N(r) \cap N'(r)$ and $i \notin N(r) \cup N'(r)$. For cases (1) and (2), we have

$$\|(\mathbf{A}'_i - \mathbf{A}_i)\mathbf{W}^{(A)}\|^2 = \|\mathbf{e}_r^T \mathbf{W}^{(A)}\|^2 \overset{(a)}{=} 1,$$

where $(a)$ is due to the fact that $\mathbf{W}^{(A)}$ is row-normalized. By the bounded out-degree assumption, we know that $\max(|N(r)|, |N'(r)|) \leq D$. Thus, the worst case upper bound of $\|[\mathbf{A}'\mathbf{W}^{(A)}]_{\backslash r} - [\mathbf{A}\mathbf{W}^{(A)}]_{\backslash r}\|_F^2$ arises when $N(r)$ and $N'(r)$ are disjoint, which results in a contribution of $2D$ for cases (1) and (2). As a result, we have

$$\|[\mathbf{A}'\mathbf{W}^{(A)}]_{\backslash r} - [\mathbf{A}\mathbf{W}^{(A)}]_{\backslash r}\|_F^2 = \sum_{i \in [n] \backslash \{r\}} \|(\mathbf{A}'_i - \mathbf{A}_i)\mathbf{W}^{(A)}\|^2 \leq 2D.$$

This implies a sensitivity bound $\sqrt{2D}$. As a result, using the Gaussian noise mechanism with standard deviation $s$ leads to the term $\frac{2D\alpha}{2s^2}$ in the divergence bound. Finally, since the entire $\mathbf{W}^{(A)}$ is DP by the assumption, applying the standard DP composition theorem [40] completes the proof. $\square$

Regarding the assumption used, it can met by applying a standard DP-optimizer with an group size $D + 1$, where $\gamma_1$ depends on the noise multiplier of the DP-optimizer. This is due to the fact that the out-degree is bounded by $D$, so replacing one node can affect at most $D$ neighbors.

## D Proof of Theorem 5.4

**Theorem.** *For any $\mathcal{D} \overset{N_k}{\sim} \mathcal{D}'$, let the index of the replaced node be $r$; also, let $\mathbf{Z}_{\backslash r}$ be the matrix $\mathbf{Z}$ with its $r^{th}$ row excluded, and assume that $D_\alpha((\mathbf{W}^{(A)}, \mathbf{b})\|(\mathbf{W}^{(A)}, \mathbf{b})') \leq \gamma_1$. For any $\alpha > 1$, the embedding $\mathbf{Z}$ in DPDGC satisfies $D_\alpha(\mathbf{Z}_{\backslash r} \| \mathbf{Z}'_{\backslash r}) \leq \gamma_1 + \frac{k\alpha}{2s^2}$.*

*Proof.* The proof is nearly identical to the proof for the node-GDP case (Theorem 5.3). The only difference arises when for the case $i \in [n] \backslash \{r\}$. Recall that our goal is to analyze

$$\|[\mathbf{A}'\mathbf{W}^{(A)}]_{\backslash r} - [\mathbf{A}\mathbf{W}^{(A)}]_{\backslash r}\|_F^2 = \sum_{i \in [n] \backslash \{r\}} \|(\mathbf{A}'_i - \mathbf{A}_i)\mathbf{W}^{(A)}\|^2.$$

Again, there are four possible cases for $i \in [n] \backslash \{r\}$: (1) $i \in N(r) \backslash N'(r)$; (2) $i \in N'(r) \backslash N(r)$; (3) $i \in N(r) \cap N'(r)$; and (4) $i \notin N(r) \cup N'(r)$. Clearly, for $i$ covered by cases (3) and (4), the corresponding row norm is 0 as in this case, $\mathbf{A}'_i = \mathbf{A}_i$. For cases (1) and (2), we have

$$\|(\mathbf{A}'_i - \mathbf{A}_i)\mathbf{W}^{(A)}\|^2 = \|\mathbf{e}_r^T \mathbf{W}^{(A)}\|^2 \overset{(a)}{=} 1,$$

where $(a)$ is due to the fact that $\mathbf{W}^{(A)}$ is row-normalized. By the definition of $\mathcal{D} \overset{N_k}{\sim} \mathcal{D}'$, we know that there are at most $k$ rows corresponding to cases (1) and (2):

$$|\{i \in N(r) \backslash N'(r)\}| + |\{i \in N(r)' \backslash N(r)\}| \leq k.$$

As a result, we have

$$\|[\mathbf{A}'\mathbf{W}^{(A)}]_{\backslash r} - [\mathbf{A}\mathbf{W}^{(A)}]_{\backslash r}\|_F^2 = \sum_{i \in [n] \backslash \{r\}} \|(\mathbf{A}'_i - \mathbf{A}_i)\mathbf{W}^{(A)}\|^2 \leq k.$$

This implies a sensitivity bound of $\sqrt{k}$. As a result, applying the Gaussian noise mechanism with standard deviation $s$ leads to the term $\frac{k\alpha}{2s^2}$ in the divergence bound. Finally, since the entire $\mathbf{W}^{(A)}$ is DP by assumption, applying the standard DP composition theorem [40] completes the proof. $\square$

# E  Proof of Theorem 5.1

**Theorem.** *For any $\alpha > 1$ and $\mathcal{D} \overset{N}{\sim} \mathcal{D}'$ or $\mathcal{D} \overset{N_k}{\sim} \mathcal{D}'$, assume that $D_\alpha(\mathbf{W}^{(X)}||\mathbf{W}^{(X)'}) \leq \gamma_1$ and that both $\mathbf{A}$, $\mathbf{A}'$ have bounded out-degree $D$. Let the replaced node index be $r$ and let $\mathbf{Z}_{\backslash r}$ be the matrix $\mathbf{Z}$ excluding its $r^{th}$ row. Then the embedding $\mathbf{Z}$ used in GAP satisfies $D_\alpha(\mathbf{Z}_{\backslash r}||\mathbf{Z}'_{\backslash r}) \leq \gamma_1 + \frac{4DL\alpha}{2s^2}$.*

*Proof.* We start by showing that for any $\mathcal{D} \overset{N}{\sim} \mathcal{D}'$, $D_\alpha(\mathbf{H}^{(1)}_{\backslash r}||\mathbf{H}^{(1)'}_{\backslash r}) \leq \gamma_1 + \frac{4D\alpha}{2s^2}$. By examining $\|[\mathbf{AH}]_i - [\mathbf{A}'\mathbf{H}']_i\|$ for all $i \in [n] \setminus \{r\}$, we find that there are three cases that contribute nonzero norms. Let $N(r)$ and $N'(r)$ denotes the neighbor node set of $r$ with respect to $\mathbf{A}$ and $\mathbf{A}'$, respectively. The three cases are: (1) $i \in N(r) \setminus N'(r)$, (2) $i \in N'(r) \setminus N(r)$, and (3) $i \in N(r) \cap N'(r)$. For cases (1) and (2), $\|[\mathbf{AH}]_i - [\mathbf{A}'\mathbf{H}']_i\| \leq 1$ due to $\mathbf{H}^{(0)}$ and $\mathbf{H}^{(0)'}$ being row-normalized. For case (c), we have $\|[\mathbf{AH}]_i - [\mathbf{A}'\mathbf{H}']_i\| = \|\mathbf{H}_r - \mathbf{H}'_r\| \leq 2$. Since the out-degree is upper bounded by $D$, we know that $\max(|N(r)|, |N'(r)|) \leq D$. The worst-case arises for $|N(r) \cap N'(r)| = D$. Hence we have the following worst-case upper bound

$$\|[\mathbf{AH}]_{\backslash r} - [\mathbf{A}'\mathbf{H}']_{\backslash r}\|_F^2 = \sum_{i \in [n]\setminus\{r\}} \|[\mathbf{AH}]_i - [\mathbf{A}'\mathbf{H}']_i\|^2 \leq 4D^2.$$

This leads to the term $\frac{4D\alpha}{2s^2}$ in the divergence bound. By applying Theorem B.1 and the assumption on $\mathbf{W}^{(X)}$, we can show that $D_\alpha(\mathbf{H}^{(1)}_{\backslash r}||\mathbf{H}^{(1)'}_{\backslash r}) \leq \gamma_1 + \frac{4D\alpha}{2s^2}$.

For the $L$-hop result $\mathbf{Z}_{\backslash r}$, one can apply Theorem B.1 and induction. The base case $L = 1$ is already established above. For the induction step, we have $D_\alpha(\mathbf{H}^{(L-1)}_{\backslash r}||\mathbf{H}^{(L-1)'}_{\backslash r}) \leq \gamma_1 + \frac{4D(L-1)\alpha}{2s^2}$. By applying Theorem B.1 and repeating the previous analysis, we have

$$D_\alpha(\mathbf{H}^{(L)}_{\backslash r}||\mathbf{H}^{(L)'}_{\backslash r}) \leq \gamma_1 + \frac{4D(L-1)\alpha}{2s^2} + \frac{4D\alpha}{2s^2} = \gamma_1 + \frac{4DL\alpha}{2s^2}. \tag{3}$$

Here, the key idea is that although $\mathbf{H}^{(L-1)}_r$ is not private, it is still row-normalized so that the above sensitivity analysis still applies. For the case of $\mathcal{D} \overset{N_k}{\sim} \mathcal{D}'$, the worst case scenario still arises for $|N(r) \cap N'(r)| = D$, for all $k \geq 0$. This is in fact the case for $N_0$ (i.e., $\mathbf{A} = \mathbf{A}'$). Thus, the same result holds for the case $\mathcal{D} \overset{N_k}{\sim} \mathcal{D}'$, for all $k \geq 0$. This completes the proof. $\square$

# F  RDP to DP conversion

For a given $(\alpha, \gamma)$ Rényi DP guarantee, the following conversion lemma [34–36] allows us to convert it back to a $(\varepsilon, \delta)$-DP guarantee:

**Lemma F.1.** *If $\mathcal{A}$ satisfies $(\alpha, \gamma(\alpha))$-RDP for all $\alpha > 1$, then, for any $\delta > 0$, $\mathcal{A}$ satisfies $(\varepsilon_{\mathsf{DP}}(\delta), \delta)$-DP, where*

$$\varepsilon_{\mathsf{DP}}(\delta) = \inf_{\alpha > 1} \gamma(\alpha) + \frac{\log(1/\alpha\delta)}{\alpha - 1} + \log(1 - 1/\alpha).$$

# G  Edge GDP analysis for DPDGC and GAP

**DPDGC.** In this setting, ensuring that the embedding $\mathbf{Z}$ is DP is sufficient to guarantee the overall model being GDP. We can replace the remaining DP-MLP modules in Figure 2 with standard MLP (i.e., training with a standard optimizer).

**Theorem G.1.** *For any $\alpha > 1$ and $\mathcal{D} \overset{E}{\sim} \mathcal{D}'$, assume $D_\alpha((\mathbf{W}^{(A)}, \mathbf{b})||(\mathbf{W}^{(A)}, \mathbf{b})') \leq \gamma_1$. Then the embedding $\mathbf{Z}$ of DPDGC satisfies $D_\alpha(\mathbf{Z}||\mathbf{Z}') \leq \gamma_1 + \frac{\alpha}{2s^2}$.*

*Proof.* Since there is only one replaced entry of the adjacency matrix for any $\mathcal{D} \overset{E}{\sim} \mathcal{D}'$, $\|(\mathbf{A}' - \mathbf{A})\mathbf{W}^{(A)}\|_F \leq \max_{i \in [n]} \|\mathbf{W}^{(A)}_i\|_2 = 1$ (i.e., the maximum row-norm of $\mathbf{W}^{(A)}$). This implies that the sensitivity of $\mathbf{AW}^{(A)}$ is 1. Thus, applying the Gaussian noise mechanism with standard deviation $s$ and standard DP composition rule [40] results in $D_\alpha(\mathbf{Z}||\mathbf{Z}') \leq \gamma_1 + \frac{\alpha}{2s^2}$. $\square$

Regarding the assumption, it can be met by applying a standard DP-optimizer with group size 1, where $\gamma_1$ depends on the noise multiplier of the DP-optimizer. This is due to the fact that at most one row of $\mathbf{A}$ is different for any $\mathcal{D} \overset{E}{\sim} \mathcal{D}'$. By further applying Theorem B.1, we can establish the edge GDP guarantees for DGDGC (Corollary H.1).

**GAP.** In this setting, all DP-MLP modules in Figure 2 can be replaced with a standard MLP for GAP. Thus, we only need to ensure that the non-trainable module in GAP (i.e., PMA) is DP.

**Theorem G.2.** *For any $\alpha > 1$ and $\mathcal{D} \overset{E}{\sim} \mathcal{D}'$, the embedding $\mathbf{Z}$ of GAP satisfies $D_\alpha(\mathbf{Z}||\mathbf{Z}') \leq \frac{L\alpha}{2s^2}$.*

*Proof.* The analysis is similar to that of Theorem G.1, except that we have $\mathbf{AH}$ instead of $\mathbf{AW}_A$. Following the same argument, we know that $D_\alpha(\mathbf{H}^{(1)}||\mathbf{H}^{(1)'}) \leq \frac{\alpha}{2s^2}$. Then, by the standard DP composition theorem, we arrive at the claimed result. $\qquad\square$

# H  Formal GDP guarantees for the complete DPDGC and GAP models

**Corollary H.1.** *For any $\alpha > 1$, the DPDGC model is edge $(\alpha, \gamma_1 + \frac{\alpha}{2s^2})$-GDP.*

*Proof.* This is a direct consequence of applying Theorem G.1 (for guarantees pertaining to $\mathbf{Z}$) and the DP composition theorem. $\qquad\square$

**Corollary H.2.** *Assume that the graph has bounded out-degree $D$. For any $\alpha > 1$, the DPDGC model is node $(\alpha, \gamma_1 + \gamma_2 + \frac{2D\alpha}{2s^2})$-GDP, where the remaining weights satisfy $(\alpha, \gamma_2)$-RDP.*

*Proof.* This is a direct consequence of Theorem 5.3 (for guarantees pertaining to $\mathbf{Z}_{\setminus r}$), the standard DP-optimizer result [14] and Theorem B.1 (the generalized adaptive composition theorem). $\qquad\square$

**Corollary H.3.** *For any $\alpha > 1$, the DPDGC model is $k$-neighbor $(\alpha, \gamma_1 + \gamma_2 + \frac{k\alpha}{2s^2})$-GDP, with the remaining weights are $(\alpha, \gamma_2)$-RDP.*

*Proof.* This follows by applying Theorem 5.4 (with guarantees for $\mathbf{Z}_{\setminus r}$), the standard DP-optimizer result [14], and Theorem B.1 (generalized adaptive composition theorem). $\qquad\square$

**Corollary H.4.** *For any $\alpha > 1$, the GAP model is edge $(\alpha, \frac{L\alpha}{2s^2})$-GDP.*

*Proof.* This follows by applying Theorem G.2 (guarantees of $\mathbf{Z}$) and the DP composition theorem. $\qquad\square$

**Corollary H.5.** *Assume that the underlying graph has bounded out-degree $D$. For any $\alpha > 1$, the GAP model is node or $k$-neighbor $(\alpha, \gamma_1 + \gamma_2 + \frac{4DL\alpha}{2s^2})$-GDP, while the remaining weights are $(\alpha, \gamma_2)$-RDP.*

*Proof.* This follows by applying Theorem 5.1 (with guarantees for $\mathbf{Z}_{\setminus r}$), the standard DP-optimizer result [14], and Theorem B.1 (generalized adaptive composition theorem). $\qquad\square$

# I  Discussion on inductive settings

While we focus on the main text was on the transductive setting, we note that the inductive setting is actually significantly easier from the perspective of insuring privacy. There are two different settings for inductive graph learning. We call the first scenario the "fully inductive setting". What this means is that the information pertaining to test nodes is completely unavailable during the training phase and not subject to privacy protection [3]. One can think of this as a case where the training graph and the test graph are disjoint. In this case, model DP is sufficient to guarantee prediction DP, as one can only access training data information through the model weights. As a result, the extension of DP-SGD for GNNs [15] suffices to ensure rigorous user data privacy.

We refer to the second setting as the "incrementally inductive setting". What this means is that we will use all training and test node information during inference, despite the test nodes not being used during the training phase. In this case, DP of model weight does not ensure the DP of the model

prediction. Hence, we will need our GDP guarantees to ensure user data privacy protection. Since we only need to ensure the privacy of training data, the RDP condition in Definition 4.2 no longer needs to specify which test node is to be predicted. This is due to the fact that any test node $v$ that could be the target for prediction cannot be a training node. Thus, we will not replace it in the adjacent dataset. The remainder of the GDP analysis is similar to the previous one.

## J  Practical privacy meaning of $k$ in $k$-neigbor GDP

The notion of $k$-neighbor GDP with $(\epsilon, \delta) = (0, 0)$ implies that an adversary cannot simultaneously infer information about any $k$ in-edges and $k$ out-edges in $\mathbf{A}$. As discussed in the main text, even the case $k = 1$ already provides a similar but stronger topology privacy protection than edge GDP. Although an adversary cannot infer the existence of any edges for the $k = 1$ case, they might be confident that there is one edge among two node pairs even though they cannot be certain which one it is. For instance, even though an adversary cannot individually infer whether $\mathbf{A}_{12}$ and $\mathbf{A}_{13}$ are 1 or 0, they may be confident that the event $\{\mathbf{A}_{12} = 1 \vee \mathbf{A}_{13} = 1\}$ is true. The $k = 2$ setting provides additional protection so that the adversary cannot simultaneously infer information about any $k = 2$ edges. Nevertheless, the adversary may still be confident that the event $\{\mathbf{A}_{12} = 1 \vee \mathbf{A}_{13} = 1 \vee \mathbf{A}_{14} = 1\}$ is true (in the worst case). A similar reasoning holds for general values of $k$. As a result, selecting an intermediate value of $k$ may reveal some edge information, but it allows for a trade-off between the potentially sensitive edge information and model utility. It is worth noting that in many practical scenarios, edge information is less sensitive than that of node features and labels, making this notion of privacy particularly useful. For instance, if we are satisfied with the graph structure protection from edge-level GDP, we can simply use $k = 1$ for additional protection of sensitive node features and labels, with similar graph topology privacy.

## K  Additional experimental details

**Datasets.** We test seven benchmark datasets available from either the Pytorch Geometric library [42] or prior works. These datasets include the social network **Facebook** [17, 43], citation networks **Cora** and **Pubmed** [44, 45], the Amazon co-purchase networks **Photo** and **Computers** [46], and Wikipedia networks **Squirrel** and **Chameleon** [47]. Pertinent dataset statistics can be found in Table 1. We also report the class insensitive edge homophily measure defined in [39], with value in $[0, 1]$; the value 1 indicates the strongest possible homophily. Its definition is as follows:

$$\text{homophily} = \frac{1}{C-1} \sum_{c=1}^{C} \max(0, h_c - \frac{|\{i : \mathbf{Y}_{ic} = 1\}|}{n}),$$

$$h_c = \frac{|\{(v, w) : (v, w) \in \mathcal{E} \wedge \mathbf{Y}_{vc} = \mathbf{Y}_{wc} = 1\}|}{|\mathcal{E}|}.$$

Note that $h_c$ represents the edge homophily ratio of nodes of class $c$, where the general edge homophily is defined in [52].

**The cSBM model.** We mainly follow the cSBM model as described in [19]. The cSBM model includes Gaussian random vector node features in addition to the classical SBM graph topology. For simplicity, we assume that there are $C = 2$ equally sized communities with node labels $v_i \in \{+1, -1\}$. Each node $i$ is associate with a $f$ dimensional Gaussian vector $b_i = \sqrt{\frac{\mu}{n}} v_i u + \frac{Z_i}{\sqrt{f}}$, where $n$ is the number of nodes, $u \sim N(0, I/f)$, and $Z_i \in \mathbf{R}^f$ has independent standard normal entries. The graph in cSBM is described by an adjacency matrix $\mathbf{A}$ defined as

$$\mathbf{P}(\mathbf{A}_{ij} = 1) = \begin{cases} \frac{d + \lambda\sqrt{d}}{n}, & \text{if } v_i v_j > 0 \\ \frac{d - \lambda\sqrt{d}}{n}, & \text{otherwise} \end{cases}.$$

Similar to the classical SBM, given the node labels, the edges are independent. The symbol $d$ stands for the average degree of the graph. Also, recall that $\mu$ and $\lambda$ control the strength of the information content conveyed by the node features and the graph structure, respectively.

One reason for using the cSBM to generate synthetic data is that the information-theoretic limit of the model has already been characterized in [18]. This result is summarized below.

**Theorem K.1** (Informal main result from [18]). *Assume that $n, f \to \infty$, $\frac{n}{f} \to \xi$ and $d \to \infty$. Then there exists an estimator $\hat{v}$ such that $\liminf_{n \to \infty} \frac{|\langle \hat{v}, v \rangle|}{n}$ is bounded away from 0 if and only if $\lambda^2 + \frac{\mu^2}{\xi} > 1$.*

In our experiment, we set $n = 10,000, f = 200$, so that $\xi = 50$. We vary $\mu$ and $\lambda$ along the arc $\lambda^2 + \mu^2/\xi = 1 + \epsilon$, for some $\epsilon > 0$, to ensure that we are within the allowed parameter regime. We also set $\epsilon = 3.25$ in all our experiments.

**Experiment environments.** All experiments are performed on a Linux Machine with 48 cores, 376GB of RAM, and an NVIDIA Tesla P100 GPU with 12GB of GPU memory. We use PyTorch Geometric[3] [42] for graph-related operations and models, autodp[4] for privacy accounting and Opacus[5] [53] for the DP-optimizer. Our code is developed based on the GAP repository[6] [17] and follows a similar experimental pipeline.

**Hyperparameters.** For all methods, we set the hidden dimension to $64$, and use SeLU [54] as the nonlinear activation function. The learning rate is set to $10^{-3}$, and do not decay the weights. Training involves 100 epochs for both pretraining and classifier modules. We use a dropout $0.5$ for nonprivate and edge GDP experiments and no dropout for the node GDP and $k$-neighbor GDP experiments. For DPDGC, we find that row-normalizing $\mathbf{W}^{(A)}$ to $10^{-8}$ and reducing $s$ accordingly gives better utility in practice (see Algorithm 2, where we set $c = 10^{-8}$). Following the analysis of DPDGC, we know that this changes the sensitivity of $\mathbf{AW}^{(A)}$ to $\sqrt{2Dc}$ for node GDP and $\sqrt{kc}$ for $k$-neighbor GDP, respectively. Note that when $c = 1$, we recover the results of Theorem 5.3 and 5.4. For GAP, we tune the number of hops $L \in \{1, 2, 3\}$. For both DPDGC and GAP, the MLP modules have 2 layers in general, except for the $\text{MLP}_A$ and $\text{MLP}_X$ modules in DPDGC which have 1 layer. For MLP, we use 3 layers following the choice of [17]. We upper bound the out-degree by $D = 100$ for all datasets, following the default choice stated in [17]. The batch size depends on the dataset size, where we choose 256 for Facebook and Pubmed, and $64$ for the rest. Note that we train without mini-batching whenever the training does not involve a DP-optimizer and the method has better performance. For the cSBM experiments, we adopted the settings used for the Facebook dataset. We observe that DPDGC (nopriv) can have severe overfitting issues for $\phi = 0$, which causes the results to be unstable. For this particular case, we change the learning rate of the embedding module to $10^{-5}$ in order to mitigate the issue.

**DP-optimizer.** We use DP-Adam by leveraging the Opacus library. We retain the default setting for all methods.

## L   Pseudocode for GAP and DPDGC

---

[3]https://github.com/pyg-team/pytorch_geometric
[4]https://github.com/yuxiangw/autodp
[5]https://github.com/pytorch/opacus
[6]https://github.com/sisaman/GAP

---
**Algorithm 1** Training process of GAP
---
1: **Model input:** node feature $\mathbf{X}$, adjacency matrix $\mathbf{A}$, training labels $\mathbf{Y}$.
2: **Parameters:** noise std $s$, hops $L$, and max degree $D$.
3: **if** edge GDP **then**
4:     Train DP-MLP$_X$ using a standard optimizer.
5: **else**
6:     Train DP-MLP$_X$ using a DP-optimizer with group size of 1.
7: **end if**
8: $\mathbf{H} \leftarrow$ DP-MLP$_X(\mathbf{X})$.
9: $\mathbf{H}^{(0)} \leftarrow$ row-normalized($\mathbf{H}$) and cache $\mathbf{H}^{(0)}$.
10: **if** node or $k$-neighbor GDP **then**
11:     Subsample $\mathbf{A}$ so that the out-degree (column-sum) is bounded by $D$.
12: **end if**
13: **for** $1 \le i \le L$ **do**
14:     $\mathbf{H}^{(i)} \leftarrow$ row-normalized($\mathbf{H}^{(i-1)} + N(0, s^2)$).
15: **end for**
16: $\mathbf{Z} \leftarrow \|_{i=0}^{L}\mathbf{H}^{(i)}$ and cache $\mathbf{Z}$.
17: **if** edge GDP **then**
18:     Train DP-MLP$_f$ using a standard optimizer.
19: **else**
20:     Train DP-MLP$_f$ using a DP-optimizer with a group size 1.
21: **end if**
---

---
**Algorithm 2** Training process of DPDGC
---
1: **Model input:** node feature $\mathbf{X}$, adjacency matrix $\mathbf{A}$, training labels $\mathbf{Y}$.
2: **Parameters:** noise std $s$, maximum degree $D$ (for node GDP) or $k$ for $k$-level GDP, row-normalized constant $c$.
3: **if** node GDP **then**
4:     Subsample $\mathbf{A}$ so that the out-degree (column-sum) is bounded by $D$.
5: **end if**
6: **if** node GDP **then**
7:     $x = D + 1$
8: **else if** $k$-neighbor GDP **then**
9:     $x = k + 1$
10: **else**
11:     $x = 1$
12: **end if**
13: $(\mathbf{W}^{(A)}, \mathbf{b}) \leftarrow$ Train DP-Emb using a DP-optimizer with group size $x$ and constrain $\mathbf{W}^{(A)}$ to be row-normalized to $c$.
14: $s \leftarrow c \times s$.
15: $\mathbf{Z} \leftarrow$ row-normalized($\mathbf{A}\mathbf{W}^{(A)} + N(0, s^2) + \mathbf{b}$) and cache $\mathbf{Z}$.
16: **if** edge GDP **then**
17:     Train the remaining modules using a standard optimizer.
18: **else**
19:     Train the remaining modules using a DP-optimizer with group size 1.
20: **end if**
---

