# OpenReview forum: "Differentially Private Decoupled Graph Convolutions for Multigranular Topology Protection"
_NeurIPS.cc/2023/Conference — NeurIPS 2023 poster_

### Official Review · Reviewer_fPNi · 2023-07-01

**Soundness:** 3 good
**Presentation:** 3 good
**Contribution:** 3 good
**Rating:** 5
**Confidence:** 5

**Summary:**

Graph neural networks have privacy leakage in both their topology information and node attribute information. This paper proposes a differential privacy framework to protect both graph topology and node attributes. A model that decouples graph convolution and node attribute embedding is proposed.

**Strengths:**

A graph differential privacy (GDP) framework is proposed for GNN models. Theoretical GDP guarantees are provided.

**Weaknesses:**

1. A weakness of using the differential privacy (DP) metric is the significant deterioration of utility (in this paper, it is the node classification accuracy) for even a very generous privacy budget. As seen in Table 3 and Figure 3, \epsilon=16 has to be set to achieve reasonable accuracy (except for the simplest case of edge-level privacy). However, even with such a generous budget, the test accuracy drops significantly compared to the non-private case. One should question if DP is indeed the proper framework to use in GNN (despite its popularity in database privacy). For example, there are frameworks on *inference* privacy that specifically protect certain private attributes instead of the full "data" (graph topology  + features), which are more applicable in practice.

2. By decoupling the graph adjacency information A from the node attributes X, the model can no longer benefit from graph aggregation and local node processing. This also explains why the proposed model does not perform well on homophily datasets.

**Questions:**

1. It was not immediately clear by the end of Section 5 that the output of DP-MLP_W is also designed to be DP, hence the overall framework is DP due to the composition theorem. It caused some confusion for me and I would appreciate if this point is emphasized since DP-MLP_W is never discussed in detail throughout the paper.

2. How is the MLP in Table 3 trained to achieve GDP? If the MLP is GDP and protects graph information (i.e., using the individual outputs from the MLP applied to individual nodes, one cannot easily infer if there are edges between them), why does it perform significantly better than DPDGC or DP-SAGE?

3. On the heterophily datasets, the reduction in test accuracy is very significant compared to the non-private scenario. What is causing this? A detailed discussion should be added.

4. How tight are the bounds in the theoretical results in Section 5 and are these used directly in the DPDGC model or is the model tuned instead based on an empirical estimate of its GDP?

5. What is the computational complexity compared to baselines? Is a distributed implementation using message passing possible?


**Limitations:**

Yes

---

> ### Author Rebuttal · Authors · 2023-08-08
>
> We thank reviewer fPNi for their thoughtful feedback, comments, and positive assessments of our work. All questions are addressed below.
>
> - W1: ``A weakness of using the differential privacy (DP) metric is the significant deterioration of utility … for even a very generous privacy budget. … One should question if DP is indeed the proper framework to use in GNN (despite its popularity in database privacy).``
>
> This is indeed an interesting comment. Please see G3 of our general response.
>
> - W2: ``By decoupling the graph adjacency information A from the node attributes X, the model can no longer benefit from graph aggregation and local node processing.``
>
> Thank you for pointing this out. We have indeed discussed this issue and described it in our “limitation” section in the Appendix (lines 506-512). The key contributions of our work are to introduce new differential privacy criteria for graph learning tasks and establish theoretical privacy guarantees for different learning settings. Decoupling the graph adjacency matrix allowed us to better control the privacy-utility trade-offs and it was fundamental for motivating partial topology privacy concepts such as $k$-neighbor-level GDP. Note that traditional message-passing designs mostly focus on the utility aspect of GNNs: Whether there exists a GNN design that can achieve our $k$-neighbor-level GDP and offer similar utility as standard message-passing GNNs remains a challenging open problem. It is also worth noting that LINKX [38] also adopts a similar decoupled structure and empirically demonstrates strong (nonprivate) performance on heterophilic datasets. Nevertheless, that work does not study privacy-related features of decoupling.
> - Q1: ``It was not immediately clear by the end of Section 5 that the output of DP-MLP_W is also designed to be DP, …``
>
> We apologize for the confusion. Once we obtain the cached intermediate embedding $Z$, the remaining DP-MLP modules are trained with DP-SGD in an end-to-end fashion. We will clarify this point in our revision.
>
> - Q2: ``Questions about DP-MLP``
>
> Please see G1 in our general response.
>
> - Q3: `` Why on the heterophily datasets, the reduction in test accuracy is very significant compared to the non-private scenario.``
>
> This is a good question. We conjecture that the beneficial information in Squirrel and Chameleon datasets is relatively delicate. That is, most node embeddings are close to the classification boundaries of GNNs even if they are correctly classified in the non-private case. Thus, adding noise during the computation of $Z$ embedding may greatly obfuscate these beneficial signals. It is unclear at this point why this phenomenon happens, and we hope to further investigate it in the future.
>
> - Q4: ``How tight are the bounds in the theoretical results in Section 5 and are these used directly in the DPDGC model?``
>
> Note that DP is defined with respect to the worst case scenario, and following our proof in Section 5, one can easily construct a worst case pair ($(X, A)$ and $(X’, A’)$) such that the bounds are tight, implying our bounds are worst-case optimal. On the other hand, we use a tight upper bound on sensitivity in our experiment to obtain a practical GDP guarantee. As stated in Appendix I (line 677), we use autodp for privacy accounting (which adopts privacy amplification via subsampling and compositions via Renyi DP). We note that the conversion from RDP to approximate DP  (Lemma F.1) is known to be nearly optimal and has been used in most practical DP frameworks.
>
> - Q5: ``What is the computational complexity compared to baselines?``
>
> The computational complexity of DPDGC and GAP are roughly the same. Note that the bottleneck of the computational complexity is the operation $AX$, $AH$ (GAP), or $AW$ (DPDGC). We set the (hidden) dimension of $W$ to be 64 (Appendix I, line 680), which is usually smaller than the feature dimension of $X$. As a result, the computational complexity of DPDGC, GAP, and the other GNN baselines are similar. Note that the implementation of DPDGC and message-passing in Pytorch Geometric already uses sparse tensor/matrix multiplication to speed up the computation of $AX$, $AH$, or $AW$. Furthermore, similar to LINKX [38], DPDGC by default can support the arbitrary size of mini-batch training and is thus scalable. Please see G2 in our general response for further discussion.
>
> Please feel free to let us know if there are follow-up questions. We will try our best to address them in a timely manner.

---

> > ### Comment · Reviewer_fPNi · 2023-08-12
> >
> > Thank you for your clarifications. I appreciate the authors' efforts in responding to the reviews. I have no further questions.

---

> > > ### Author Response · Authors · 2023-08-14
> > >
> > > Thank you for the acknowledgment!

---

### Official Review · Reviewer_ssFv · 2023-07-04

**Soundness:** 2 fair
**Presentation:** 2 fair
**Contribution:** 2 fair
**Rating:** 4
**Confidence:** 5

**Summary:**

This paper introduces a new framework called Differentially Private Decoupled Graph Convolutions (DPDGC) for graph learning settings that ensures both provably private model parameters and predictions. The framework is designed to protect sensitive user information and interactions in graph-structured data. The authors highlight the limitations of standard Differential Privacy techniques in graph learning settings and propose a novel notion of relaxed node-level data adjacency to establish guarantees for different degrees of graph topology privacy. The paper also includes an analysis of the framework and its performance compared to existing methods.

**Strengths:**

1. This paper focuses on the important problem of graph differential privacy, which is critical in protecting sensitive user information and interactions in graph-structured data.
2. This paper conducts experimental evaluation on seven node classification benchmarking datasets.

**Weaknesses:**

1. The paper's presentation is difficult to follow, which may make it challenging for readers, especially for those who do not have strong background knowledge on this topic hard to understand the proposed method and its contribution.
2. The proposed method has poor performance, which is reflected in the experimental results presented in the paper.
3. Some relevant literature has not been cited in the paper, which could suggest that the authors have not conducted a thorough review of the existing research in this area.

**Questions:**

1. How can one determine the appropriate value of K for a given dataset, in order to achieve meaningful results considering both privacy protection and utility?
2. In the case of the Pubmed and Cora datasets, why do the results of graph-based machine learning methods remain unchanged across different values of K, and what implications does this have for the use of these datasets in research?


**Limitations:**

1. The paper's presentation could be improved to make it easier for the reader to follow the content and understand the proposed method.
2. The utility of the proposed method is limited, as the privacy protection achieved with a privacy budget of eps=16 is too weak to be meaningful for many datasets. Furthermore, even with this level of privacy protection, the performance of the proposed method is significantly worse than that of non-private baseline models on most datasets, indicating poor utility.
3. The paper could benefit from a more comprehensive review of related works, as some relevant studies are not cited or discussed in the text, including [1].

[1] Zhang, Q., Ma, J., Lou, J., Yang, C., & Xiong, L. (2022). Towards Training Graph Neural Networks with Node-Level Differential Privacy. arXiv preprint arXiv:2210.04442.

---

> ### Author Rebuttal · Authors · 2023-08-08
>
> We thank reviewer ssFv for their comments. We addressed all questions below.
>
> - W1: `` The paper's presentation is difficult to follow.``
>
> We appreciate the reviewer’s comments regarding readability. We spend significant efforts to find a good order of exposition and lengths of explanations under page limit constraints, but there is clearly always room for improvement. Nevertheless, we would like to point out that all other reviewers mentioned that our presentation is good (score 3). Moreover, Reviewer XfoH even explicitly mentioned that `` The problem and contributions discussed in the paper are clearly mentioned and the illustrations do a good job of conveying them.`` Reviewer tLab also mentions that ``The authors have effectively conveyed complex concepts and ideas in a well-structured manner.`` and put it as one of the strengths of our paper. We are already working on improving the presentation of our paper even further.
>
> - W2: `` The proposed method has poor performance.``
>
> We respectfully disagree with this comment. Note that our proposed method outperforms or matches the prior state-of-the-art method GAP and all the other baselines in almost all cases. Even for a few cases for which DP-MLP has the best performance, our approach is still competitive with the other DP-GNN baselines. Note that it is reasonable that in some cases, DP-MLP can outperform DP-GNNs. See G1 in our general response for further discussion and examples.
>
> In summary, we would like to emphasize again that our DPDGC model already has the overall best performance compared to other DP-GNN baselines. It is important to bear in mind how difficult it is in practice to ensure provable privacy guarantees for graph-based learners with satisfied utility.
>
> - W3: ``Some relevant literature has not been cited in the paper, which could suggest that the authors have not conducted a thorough review of the existing research in this area.``
>
> We thank reviewer ssFv for bringing to our attention the paper of Zhang et al. [ref 1]. We will include it in the related work section of our revision. However, we would like to point out that [ref 1] considers the case that the training graph and test graph are disjoint. Hence, they only need to ensure GNN weights are DP. Their method cannot be extended to the more challenging setting where training nodes are reused for inference, as is the case of our work and the common node classification scenario. See page 7 in [ref 1]. As a result, we cannot compare [ref 1] with GAP and our approach. Furthermore, their method can be viewed as a privatized version of APPNP [ref 2], which is unable to learn well on heterophilic datasets [ref 3]. In contrast, our DPDGC can work well on heterophilic datasets.
>
> - Q1: `` How can one determine the appropriate value of K for a given dataset?``
>
> One should treat the parameter $k$ in our $k$-neighbor-level GDP similar to $(\epsilon,\delta)$ as in approximate DP (see our discussion in lines 348-352); it serves as a design parameter specifically tailored to graph datasets. The appropriate value of $k$ can be determined by considering the level of sensitivity in revealing a portion of edge information. In practical implementations, the model holder (e.g., the server) should proactively choose privacy parameters $k$, $\epsilon$, and $\delta$ in compliance with regulations or users' agreements. While we demonstrate the utility-privacy trade-off for different choices of $k, \epsilon, \delta$, the final choice should be made by the practitioners in the field.
>
> - Q2: `` In the case of the Pubmed and Cora datasets, why do the results of graph-based machine learning methods remain unchanged across different values of K, and what implications does this have for the use of these datasets in research?``
>
> Note that GAP and the other DP-GNN baselines are unaffected by the choice of $k$, as already discussed in Section 5. The main reason behind this finding is their “coupling” design, i.e., the use of the product of $A$ and (a function of) $X$. We also give a detailed discussion and simple example explaining why prior DP-GNN models are not affected by the choice of $k$, see lines 244-251. On the other hand, we would like to point out that our DPDGC ***does offer different performance*** for different choices of $k$. Please check our Table 3 and Figure 3 for details. Also check our updated Table in the general response.
>
> - L1: `` The utility of the proposed method is limited, as the privacy protection achieved with a privacy budget of eps=16 is too weak ...``
>
> We would like to point out that our method already outperforms the state-of-the-art DP-GNN method, GAP, across different datasets and privacy parameter settings. We also want to point out that the authors of GAP also choose $\epsilon=16$ to demonstrate the utility of DP-GNNs for node-level DP experiments. While we agree that $\epsilon=16$ may be too weak in practice, we are not aware of any other methods that outperform our DPGDC and we do test for different $\epsilon \in \\{ 1,2,4,8,16 \\}$ in Figure 3. In fact, our novel $k$-neighbor-level GDP definition partially addresses the issue that node-level GDP with low $\epsilon$ results in poor utility. We hope the reviewer ssFv can appreciate the challenging nature of the problem of graph privacy and our contribution given the pointers above.
>
> Please feel free to let us know if there are follow-up questions. We will try our best to address them in a timely manner.
>
> ### Reference
>
> [ref 1] Towards Training Graph Neural Networks with Node-Level Differential Privacy. Zhang et al. arXiv preprint arXiv:2210.04442.
>
> [ref 2] Predict then Propagate: Graph Neural Networks meet Personalized PageRank. Gasteiger et al. ICLR 2019.
>
> [ref 3] Adaptive Universal Generalized PageRank Graph Neural Network. Chien et al. ICLR 2021.

---

> > ### Comment · Reviewer_ssFv · 2023-08-14
> >
> > Thanks for the clarifications. It's worth noting that there's relevant literature missing, specifically reference [1]:
> >
> > [1] Epasto, Alessandro, et al. "Differentially Private Graph Learning via Sensitivity-Bounded Personalized PageRank." Advances in Neural Information Processing Systems 35 (2022): 22617-22627.

---

> > > ### Author Response · Authors · 2023-08-14
> > > **About the additional reference**
> > >
> > > We thank reviewer ssFv for providing the additional reference [ref 4]. However, we would like to emphasize that [ref 4] is not about DP-GNN but specifically the DP PageRank algorithm. We believe that there can be much more literature about DP graph algorithms. Due to the space limitation, we choose to focus our discussion on DP-GNNs in the related work section, which is the most relevant to our manuscript. Note that neither our DPDGC nor discussed DP-GNN baselines leverage PageRank algorithms. The only exception is the reference Zhang et al. [ref 1] provided by the reviewer ssFv, where their work is a privatized version of APPNP and thus relevant to [ref 4]. We will try to include [ref 4] along with [ref 1] if there is still space in our revision.
> > >
> > > ### Reference
> > >
> > > [ref 1] Towards Training Graph Neural Networks with Node-Level Differential Privacy. Zhang et al. arXiv preprint arXiv:2210.04442.
> > >
> > > [ref 4] Epasto, Alessandro, et al. "Differentially Private Graph Learning via Sensitivity-Bounded Personalized PageRank." Advances in Neural Information Processing Systems 35 (2022): 22617-22627.

---

### Official Review · Reviewer_tLab · 2023-07-07

**Soundness:** 2 fair
**Presentation:** 3 good
**Contribution:** 2 fair
**Rating:** 4
**Confidence:** 2

**Summary:**

The paper presents a well-written and easily understandable framework called Graph Differential Privacy (GDP) tailored for graph learning methods. The proposed framework aims to address the privacy challenges associated with GNNs by ensuring both model parameter and prediction privacy. The authors introduce the Differentially Private Decoupled Graph Convolution (DPDGC) model, which offers superior privacy-utility trade-offs compared to existing approaches. The paper includes theoretical analysis that provides a solid foundation for the proposed model. The authors evaluate the DPDGC and compare it with existing differentially DP-GNN methods, as well as non-private models. By achieving SOTA performance, the experimental results validate the effectiveness and utility of the proposed DPDGC model in graph learning tasks.


**Strengths:**

1. The paper is its clarity and coherence. The authors have effectively conveyed complex concepts and ideas in a well-structured manner.
2. The theoretical analysis provided in the paper adds good value to the research, supporting the proposed model and enhancing its credibility.
3. The incorporation of the DPDGC model, which leverages decoupled graph convolution, achieves SOTA result.


**Weaknesses:**

1. Lack of comparison with other methods: The paper focuses primarily on comparing the performance of the proposed GDP-based methods (including DPDGC) against other differentially private graph learning methods. However, it would be beneficial to include a comparison with SOTA non-private graph learning methods to better understand the tradeoffs between privacy and utility.

2. Limited evaluation on larger and more diverse datasets: The experimental evaluation of the proposed methods is conducted on a relatively small set of benchmark datasets. The generalizability and scalability of the methods to larger and more diverse datasets are not extensively explored. Including a broader range of datasets would provide a more comprehensive evaluation of the proposed methods' performance and generalizability.

3. Lack of detailed analysis on privacy guarantees: While the paper mentions the privacy guarantees provided by the GDP framework and the DPDGC model, the detailed analysis of these guarantees is not thoroughly discussed. Providing more in-depth analysis, proofs, and discussions of the privacy guarantees would strengthen the paper's claims about the privacy properties of the proposed methods.

4. Limited exploration of alternative privacy mechanisms: The paper primarily focuses on GDP as the privacy framework and DPDGC as the corresponding graph learning model. However, there are various other privacy mechanisms and techniques available in the field of differential privacy.


**Questions:**

See Weaknesses

**Limitations:**

The authors do not adequately acknowledge potential limitations in their work, which may indicate a lack of comprehensive understanding of the challenges and constraints associated with the proposed framework.

---

> ### Author Rebuttal · Authors · 2023-08-08
>
> We thank reviewer tLab for their thoughtful feedback and comments. We addressed all questions below.
>
> - W1: `` Lack of comparison with other methods: ... include a comparison with SOTA non-private graph learning methods to better understand the tradeoffs between privacy and utility.``
>
> We appreciate the comment. We agree that comparing our method with SOTA ***non-private*** graph learning methods will reveal the exact price we need to pay for privacy (GNN GDP). However, we would like to point out that studying GDP for graph learning methods is very different from studying standard DP neural networks, as incorporating (modified) DP-SGD can only ensure GNN weights being DP, but not their final output predictions. A direct injection of DP noise typically leads to very poor, if not meaningless, results. See the DP-SAGE approach and the discussion in the GAP paper [19] for an example.
>
> For standard (graph-free) classification problems, one can simply adopt DPSGD during training to make any neural network model DP. That is, the model design and the privatization approach are decoupled. Unfortunately, this is not the case for graph learning as we explain in lines 41-46. When requiring both GNN weights and the output to be GDP, we have to also specifically design DPGNNs and thus not all GNN models can be privatized easily to satisfy GDP. As we also mentioned in our limitation section (Appendix A), DPDGC designs may not be the ultimate solution for DPGNNs. Due to the complex nature of the graph learning problem, one may need to jointly design the GNN and the privacy mechanism as in GAP and our DPDGC. Nevertheless, ours is currently the best GNN model that offers GDP guarantees. As a result, we only compare GNNs that can achieve GDP with the method described in our manuscript.
>
> - W2: `` Limited evaluation on larger and more diverse datasets``
>
> Thank you for this comment. While we agree that evaluating our model on larger datasets is beneficial, we believe that our tested datasets are the most diverse one known in the DPGNN literature. Note that we are the first to test DPGNNs on heterophilic datasets and various homophilic datasets. Due to the diversity of our dataset’s choice we were able (for the first time) to establish that DP-MLP can outperform all other existing DPGNNs in some cases. This phenomenon also shows the importance of our novel $k$-neighbor-level GDP definition, as it provides a new utility-graph structure privacy tradeoff.
>
> Regarding the experiments on large datasets, please check G2 in our general response.
>
> - W3: `` Lack of detailed analysis on privacy guarantees``
>
> Due to space limitations, we were only able to provide sketches of proofs in the main text (Section 5). Complete proofs were presented in  Appendices (B to H), as is standardly done with ML conference submissions. We have tried our best to explain the key steps of our analysis in Section 5, especially regarding why the “coupling” graph convolution design fails to explore the trade-off of $k$-neighbor-level GDP and utility in lines 244-251 (i.e., the sensitivity remains unchanged for different $k$). The key ingredient for establishing a GDP guarantee is to derive tight sensitivity bounds for GAP and DPDGC, which are discussed in detail in Section 5, lines 222-236 and lines 287-292. For the proof of our main theorem (i.e., Theorem 5.1, 5.3, and 5.4), please check Appendix E, C, and D, respectively. Nevertheless, we will try our best to make our sketch of proof and discussion more transparent in the revision.
>
> - W4: `` Limited exploration of alternative privacy mechanisms: The paper primarily focuses on GDP as the privacy framework and DPDGC as the corresponding graph learning model. However, there are various other privacy mechanisms and techniques available in the field of differential privacy.``
>
> We believe that our privacy definitions for graph datasets are common and appropriate in both theory and practice. It is worth noting that the notions of edge-level and node-level DP have been widely adopted and studied in numerous previous works [18,19,29]. In our work, we further extend DP definitions by introducing the concept of k-neighbor-level privacy, which offers a novel way to capture practical privacy considerations specific to graph datasets.
>
> Regarding the techniques employed to achieve GDP, we have designed a GNN architecture with small sensitivity,  compatible with the DP noise addition mechanism. Our choice of Gaussian noise as the mechanism to achieve DP is well-founded for several reasons: (1) Gaussian noise allows for tighter privacy accounting, (2) the Gaussian mechanism has been proven to achieve optimal MSE under a z-CDP constraint [ref 1] (note that z-CDP can be viewed as a variant of Renyi DP), and (3) Gaussian noise DP is easy to implement in practice. While we believe that our technique based on the Gaussian mechanism is well-suited for the task, we are also open to alternative suggestions and would be happy to implement and compare with other solutions proposed by the reviewer.
>
>  [ref 1 ] Mark Bun and Thomas Steinke, “Concentrated Differential Privacy: Simplifications, Extensions, and Lower Bounds”, TCC 2016
>
> ### Comments regarding the limitation
>
> Due to the space limit, we have put the limitation section in Appendix A. We have mentioned that current SOTA non-private GNNs on homophily datasets are still using “coupling” graph convolution designs. Thus, our proposed DPDGC may not be the ultimate solution but it is the best currently known solution under GDP constraint. We hope to further investigate whether there is a new GNN design that can have the merits of both worlds. Hence, we respectfully disagree with the comment about us not adequately acknowledging the limitations of our work.
>
> Please feel free to let us know if there are follow-up questions. We will try our best to address them in a timely manner.

---

### Official Review · Reviewer_XfoH · 2023-07-27

**Soundness:** 3 good
**Presentation:** 3 good
**Contribution:** 3 good
**Rating:** 7
**Confidence:** 3

**Summary:**

The paper introduces a differentially private GNN model that allows for different privacy requirements for node attributes and graph structure. The model decouples graph convolutions from node attributes and graph topology and provides provable privacy guarantees. Experimental results are provided to show the proposed methodology's superiority.

**Strengths:**

The problem and contributions discussed in the paper are clearly mentioned and the illustrations do a good job of conveying them. Graph differential privacy is an interesting topic and the idea of providing flexibility for node attributes and graph structure is promising.

**Weaknesses:**

Some discussion regarding the questions mentioned in the next section would add to the paper greatly.

**Questions:**

1. From line 310: "we also test (DP-)MLP and several DP-GNN baselines that can achieve GDP guarantees, including RandEdge+SAGE [29] and DP-SAGE [18] for edge and node GDP, respectively.". However, it is not clear what MLP refers to in Table 3.

2. There is a large jump in performance for the non-private setting for DPDGC but the other methods do not exhibit this behavior. Any insight into this phenomenon?

3. From line 341 - DPDGC starts to outperform GAP when privacy budget increases but lags behind when privacy budget is small. What is the typical scenario in real world situations?

4. Regarding the comment on line 330: does homophily alone decide for which datasets utility loss from privacy noise compensates graph structure information? Basically, what are the things that one has to consider before picking the right algorithm to achieve GDP?

Minor:
On line 109, what is T?
Table 3: change "none" to "non"

---

> ### Author Rebuttal · Authors · 2023-08-08
>
> We thank reviewer XfoH for their thoughtful feedback, comments, and positive assessment of our work. We addressed all questions below.
>
> - Q1: `` From line 310: "...". However, it is not clear what MLP refers to in Table 3.``
>
> We apologize for the confusion. MLP in Table 3 refers to both non-private MLP and DP-MLP, depending on whether it is in the first row of Table 3 or not (i.e., having $\epsilon>0$ or not). We will change MLP to DP-MLP in Table 3 when it is trained with DPSGD. See our pdf in the general response.
>
> - Q2: `` There is a large jump in performance for the non-private setting for DPDGC but the other methods do not exhibit this behavior. Any insight into this phenomenon?``
>
> We assume that reviewer XfoH refers to results on heterophily datasets (Squirrel, Chameleon, and Facebook). It is known in the literature that standard GNN designs (such as GCN) do not work well on heterophilic datasets. In contrast, specially designed GNNs such as LINKX [38] work much better in this setting. Thus, we conjecture that GAP (and the other DPGNN baselines) which are similar to GCN cannot perform well on heterophilic datasets even in the nonprivate setting, as described in [38] and our Table 3. As a result, adding privacy noise to these models will not result in a huge performance drop, as they already have poor performance in nonprivate settings. In contrast, since LINKX can learn well on heterophilic datasets in nonprivate settings, injecting large privacy noise into LINKX can result in a huge performance drop. Please let us know if we misinterpret your question, we will follow up in a timely manner.
>
> - Q3: `` From line 341 - DPDGC starts to outperform GAP when privacy budget increases but lags behind when privacy budget is small. What is the typical scenario in real world situations?``
>
> In practice, the privacy parameters $\epsilon,\delta$ and $k$ should be determined based on the agreement between the model holder, the users or privacy regulators. That is, the model holders and users should first determine the ***strength*** of privacy (i.e., $\epsilon, \delta$) they agree with even before training models. Given such GDP constraints, we can then identify the DPGNNs with the best utility. Our experimental results only demonstrate the privacy-utility trade-off, but do not assume any particular privacy level requirements.
>
> - Q4: `` Regarding the comment on line 330: does homophily alone decide for which datasets utility loss from privacy noise compensates graph structure information? Basically, what are the things that one has to consider before picking the right algorithm to achieve GDP?``
>
> This is an interesting question. We believe that there are multiple factors that affect this phenomenon. We conjecture that both homophily level and edge density are two good indicators. Our conjecture is based on the analysis of the Stochastic Block Model (SBM) with two clusters [ref 1]. Consider a graph that has two even-size clusters ($n/2$). “In-cluster” edges are sampled i.i.d. from Bernoulli distribution with probability $p=a \log(n)/n$. “Intra-cluster” edges are sampled i.i.d. from Bernoulli distribution with probability $q=b \log(n)/n $ for some non-negative reals $a,b$. It is known from the literature that the fundamental limit of exact recovery (i.e. recover the clusters with high probability) is $|\sqrt{a}-\sqrt{b}|>\sqrt{2}$ [ref 1]. Interestingly, this simplified case reveals two facts about how strong the graph information alone is: 1) when the homophily measure is close to $1$ or $0$ (i.e. |p-q| is large, where $p$ and $q$ are the edge density of in-cluster and intra-cluster edges respectively), the graph structure has strong information about the labels (clusters). 2) one needs edge density to be large enough. In the case of SBM, it should be $\Omega(\log(n)/n)$. If the edge density is too low (i.e., there are only n/2 edges), then it is impossible to achieve exact recovery even if $q=0$ as the graph is not even connected.
> Indeed, this only characterizes the strength of the graph information in a simplified case (2 clusters SBM). In real-world data, we also have node features and training node labels to learn from. We refer you to [ref 2] for a study of GNN utility on a generalized version of SBM, where both node features and training node labels are considered. Nevertheless, the privacy aspect of GNNs was not considered in [ref 2]. We also hope to further study the question of “when will the benefits of graph information compensate the price to privatize themselves.”
>
> Please feel free to let us know if there are follow-up questions. We will try our best to address them in a timely manner.
>
> ### References
>
> [ref 1] Community detection and stochastic block models: recent developments, Emmanuel Abbe, JMLR 2017.
>
> [ref 2] Graph Convolution for Semi-Supervised Classification: Improved Linear Separability and Out-of-Distribution Generalization, Baranwal et al., ICML 2021

---

> > ### Comment · Reviewer_XfoH · 2023-08-13
> > **Thank you for the thorough response**
> >
> > Changed the score to 7

---

> > > ### Author Response · Authors · 2023-08-14
> > >
> > > Thank you for the positive feedback and raising the score! We really appreciate the fruitful discussion with reviewer XfoH.

---

### Official Review · Reviewer_WwFs · 2023-07-27

**Soundness:** 3 good
**Presentation:** 3 good
**Contribution:** 2 fair
**Rating:** 4
**Confidence:** 3

**Summary:**

The authors introduce a new model for Graph Differential Privacy (GDP) that ensures a parametric level of topological privacy through decoupling of the graph convolution mechanism (i.e., preventing direct neighborhood aggregation of features = standard $AXW$ aggregation). The key definition is that up to $k$ in- and out-neighbours of a randomly selected single node $r$ can be modified to create  a new adjacency matrix. This makes this a hybrid between edge ($k=1) and node ($k=n) GDP, the parameter $k$ allows a tradeoff between  topology privacy and accuracy. All GNN training is done using this modified graph $D’$ to create the model parameters for inference. The main contribution of the paper is three-fold 1)  proving theoretical bounds on the Differential Privacy (DP) state of the art DPGNN called GAP [19] 2) Intuitions from analyzing the DP weakness of GAP to develop a novel Differentially Private Decoupled Graph Convolution (DPDGC) model, which benefits from decoupling graph convolution while providing GDP guarantees 3) theoretical bounds on the DP of their proposed DPDGC. The key intuition is that GAP has greater privacy leakage because they compute $A'H'$ where both adjacency matrix and features change. Motivated by this the authors propose DPDGC in which the $A'H'$ product is avoided thus providing more privacy than GAP.

**Strengths:**

Sound theoretical analysis of the two GDP models

Theoretical analysis of GAP i.e., the presence of the $A'H'$ product which is shown from Theorem 1 to be a contributing factor to the DP limitation of GAP. This leads to their derivation of DPGDC which applies a DP-MLP to adjacency matrix A using a non-linear operation on $AW^{(A)}$  to create the adjacency matrix embedding $Z$, where $W_A$ are fixed model weights. This is opposed to GAP which computes $AH$ for the $Z$ embeddings.  By ensuring $W^{(A)}$ is DP, they only need to look at $A'W^{(A)}$ versus $A'H'$ in GAP.




**Weaknesses:**


A big portion of the paper is spent defining the problem and setting up conventions for future Differential Privacy studies to be more suitable to the GNN field/setting.

The novelty of the decoupling method is not very convincing. The idea of decoupling is previously found in this paper [1]"Large Scale Learning on Non-Homophilous Graphs: New Benchmarks and Strong Simple Methods", D. Lim, F. Hohme et. al, Neurips 2021, which defines a method called LINKX for non-homophilous graphs. LINKX separately embeds the adjacency $A$ to $h^A$ and the features $X$ into $h^X$ before mixing adjacency and feature information. The decoupling method proposed here seems to be a modification of the LINKX strategy.

From Table 3, the proposed DPDGC model works well compared to the other differential privacy models when the graphs are heterophilic.  This is not surprising because the decoupling idea is known to be beneficial for heterophilic graphs [1]. However, the other baseline methods work better in a homophily setting (columns in the right side of the table) because they are not specifically designed for that kind of setting. I would expect a more adaptive method that works in both settings to be more convinced about the utility of this method.

Also from Table 3, simple MLP outperforms DPDGC in accuracy for higher $k$ on many datasets, especially homophilic ones. As pointed out by the authors themselves, protecting the graph information (higher privacy requirements) quite drastically reduces the utility of these decoupling methods. So the practical utility of this seems questionable.

**Questions:**

What is the practical meaning of high-$k$ topology protection? in other words what are the additional security benefits obtained by going from $k$ to $k+1$. Its not clear to what is the marginal utility of increasing $k$ in terms of the increasing cost to an adversary who wants to break privacy. This is an important question as varying $k$ is the major difference between this work and GAP.

How well will the decoupling method DPDGC work on large scale homophilous graphs with moderately high privacy budgets. I would like to see more extensive experimental results to justify the cost of decoupling versus simple MLP methods.

Fig 1 is hard to understand (contextualize) with a lot of undefined terms (e.g., k-neighbor level adjacency) especially as it comes so early in the paper. Consider moving to later or defining better in-text.

Def 4.3 seems ambiguous: $k$ entries of $A_{rj}$ and $A_{lj}$ are modified but what is the size of $ j + l $( Is $j+l=k$ or $2k$?). Text only says “some” $j$ and $l$.

There are multiple variations of row normalization, are you doing Euclidean norm normalization of each row, it is not clear from the text and obviously it makes a big difference in the proof as it is known that Euclidean row normalization dampens the effect of outliers [1] “Sign and rank covariance matrices” J. of Statistical Planning and Inference, Dec. 2000.


**Limitations:**

The authors have addressed the limitations of this work in the appendix. "we do not believe that the current DPDGC model is the ultimate solution for GDP-aware graph learning methods. To support this claim, we note that the nonprivate state of-the-art performance for learning on large-scale homophilic graphs is achieved by standard graph convolution models [47, 48].

The authors have stated that the proposed topic doesn't have any negative societal impacts, instead GDP can potentially protect user data and is beneficial. However this is a generic statement and needs to be proved (to what extent will  breaking GDP of a graph impact individual users?)

---

> ### Author Rebuttal · Authors · 2023-08-08
>
> We thank reviewer WwFs for their thoughtful feedback and comments. We addressed all questions below.
>
> - W1: `` A big portion of the paper is spent defining the problem and setting up conventions for future Differential Privacy studies to be more suitable to the GNN field/setting.``
>
> We believe this is in fact one of our most important contributions. Note that we need a rigorous definition and proof for GDP to truly ensure the privacy of graph datasets. As first pointed out by the authors of [19], merely ensuring GNN weight DP is insufficient to protect the privacy of graph datasets. We are the first to formally define GDP and lay rigorous theoretical foundations for future DP GNN studies.
>
> - W2: `` The novelty of DPDGC with respect to LINKX``
>
> Please see G4 in our general response.
>
> - W3: “DPDGC only works well on heterophilic graphs but not homophilic graphs”
>
> We agree that part of the reason that DPDGC works well on heterophilic graphs is due to the same reason LINKX works well on heterophilic graphs. However, we would like to emphasize that our contribution is to propose the privatization design of DPDGC and identify the theoretical privacy benefits of decoupled graph convolution design. Note that ensuring the GDP guarantee is not trivial, as a direct extension of DP-SGD training to GNN does not work (see lines 41-46). GAP is the only (and prior state-of-the-art) DP-GNNs that satisfies GDP guarantees (our Corollary H.4 and H.5) but does not work for heterophilic graph datasets as demonstrated in Table 3. Our DPDGC is currently the only method that can achieve nontrivial performance on heterophilic graph datasets with GDP guarantees.
>
> We agree that DPDGC is only on-par with GAP under the node-level GDP setting for homophilic datasets. Nevertheless, DPDGC still significantly outperforms GAP in the  $k$-neighbor-level GDP setting on three out of four homophilic datasets. While DP-MLP can outperform all DPGNNs in certain scenarios, we conjecture this to be the case due to the benefits of graph information not being able to compensate for the utility loss induced by privacy noise that protects the graph information (line 330). Please check our general response G1 for more information. Still, we agree that there should be a more adaptive design of DPGNNs that works across all settings and datasets, which we also mention in our limitation section (Appendix A). Such a design appears hard to find and work on this problem is ongoing.
>
> - W4: `` simple MLP outperforms DPDGC in accuracy for higher $k$ on many datasets``
>
> Please see G1 in our general response.
>
> - Q1: `` What is the practical meaning of high-$k$ topology protection?``
>
> This is an excellent question. One can think that the parameter choice $k=1$ provides roughly the same privacy protection on $A$ as in edge-level GDP (albeit $k$-neighbor-level GDP also protects the node features and labels). For general $k$, this implies that the adversary cannot simultaneously infer the existence of $k$ neighbors for each node. Let's consider the simplest case of $k=1$: according to Definition 4.4, the GDP algorithm output prevents the adversary from inferring the existence of any edge, even with access to the remaining n-1 nodes and their edges. However, the adversary may still be confident that there is a true edge between certain node pairs. For instance, even though the adversary cannot individually infer whether $A_{12}$ or $A_{13}$ are $1$ or $0$, they might be confident that $A_{12}=1$ or $A_{13}=1$ in the worst case. The case $k=2$ additionally protects this case but the adversary might be confident that $A_{12}=1$ or $ A_{13}=1$ or $A_{14}=1$ in the worst case. A similar explanation holds for general $k$.
> Selecting an intermediate value of $k$ (where $1 < k < n$) may reveal some portion of edge information, but it represents a trade-off between the potentially sensitive edge information and utility. It's worth noting that in many practical scenarios, edge information can be less sensitive than node features, making this notion of privacy particularly useful in various applications. For instance, if we are satisfied with the graph structure protection from edge-level GDP, we can simply use $k=1$ for additionally protecting sensitive node features and labels with a similar graph structure privacy.
> Please let us know if further clarification is needed, we are happy to provide a more intuitive explanation.
> - Q2: ``How well will the decoupling method DPDGC work on large scale homophilous graphs?``
>
> This is a good question. Please check G2 in our general response.
>
> - Q3: ``Fig 1 hard to understand, consider to move it later``
>
> Thank you for the suggestion. We will either try to make it more clear or move it to the later part of the paper as suggested.
>
> - Q4: `` Def 4.3 seems ambiguous``
>
> Sorry for the confusion. We meant to say that the total number of replaced entries is $2k$ ($k$ for in-edges and $k$ for out-edges). We will make this clear in our revision.
>
> - Q5: ``question about row normalization``
>
> We apologize for the confusion. We mean an $\ell_2$ norm (Euclidean) row normalization. That is, for a matrix $H\in \mathbb{R}^{n\times d}$, we normalize each row of $H$ to have  $\ell_2$ norm equal to one (i.e., $||H_i||_2 = 1$ for all $i\in [n]$). We will make this clear in our revision.
>
> It is interesting to investigate the outlier effect mentioned by the reviewer WwFs as a future direction. However, in this work, we merely focus on the privacy aspect and this future direction is out of the scope. We thank reviewer WwFs for this intriguing comment.
>
> Please feel free to let us know if there are follow-up questions. We will try our best to address them in a timely manner.

---

> > ### Comment · Reviewer_WwFs · 2023-08-13
> >
> > Thank you for the clarifications. No further questions as of now.

---

> > > ### Author Response · Authors · 2023-08-14
> > >
> > > Thank you for the notification and response!

---

### Author Rebuttal · Authors · 2023-08-08

We appreciate the time and effort of all reviewers and the AC. The reviews indeed provided helpful feedback. In this general response, we elaborate on some common questions raised by the reviewers (G1, G2). We also highlight some thoughtful questions and comments that initiate valuable discussions (G3, G4).

- G1: ``Why simple DP-MLP has the best performance in some cases and how is it trained`` (Reviewer fPNi, WwFs)

Note that the DP-MLP model does not leverage the graph structure information $A$. Hence, it does not need to pay an extra privacy price for protecting the graph structure. DP-MLP is simply trained with DP-SGD to achieve the same $(\epsilon,\delta)$ guarantees from the DP definition. Roughly speaking, DP-MLP can spend the whole privacy budget $(\epsilon,\delta)$ on DP-SGD training. However, both GAP and DPDGC need to spend some privacy budget to account for protecting $A$, such as the PMA module and multi-step training. As a result, DP-MLP requires less noise during training compared to GAP and DPDGC. Hence, DP-MLP might outperform GDP-GNNs such as our DPDGC when the benefit of graph structure information is insufficient to compensate for the additional noise needed for a GDP guarantee.

Consider an extreme example, where $X = Y$ and $A$ is purely random (i.e., generated from Erdos-Renyi random graph model with edge probability 1/2). Clearly, introducing $A$ will not benefit the node classification accuracy. However, in order to make the graph algorithm GDP, we still need to pay an additional privacy budget to “protect” the privacy of $A$. It is obvious that DP-MLP will outperform any possible DP-GNNs in this case. For further discussion on how to characterize graph structure information, see our response to Q4 of reviewer XfoH.

- G2: ``Performance on large homophilous graph datasets`` (Reviewer fPNi, tLab, WwFs)

We are also very interested in conducting experiments on them. Unfortunately, the Opacus library (library for DP training) currently does not support SparseTensor (see issue #579 of Opacus GitHub repository) so we are unable to run tests on larger datasets. We believe that Opacus will support SparseTensor in a future version (as indicated by issue #350 of the Opacus GitHub repository by the creators) and we are willing to test on large graph datasets such as those in the Open Graph Benchmark repository.

- G3: `` A weakness of using the differential privacy (DP) metric is the significant deterioration of utility… One should question if DP is indeed the proper framework to use in GNN.`` (Reviewer fPNi)


While we acknowledge that there might be alternative definitions of privacy that could be considered "proper" for Graph Neural Networks (GNNs), we want to emphasize that Differential Privacy (DP) has emerged as the most widely accepted and implemented  privacy standard across various fields, including machine learning, database & privacy research communities as a whole. The adoption of DP algorithms and models by prominent tech companies like Apple and Google, as well as the U.S. Census Bureau's embrace of Differential Privacy for data privacy, all further highlight its practical relevance and applicability in real-world settings. Given the widespread acceptance and deployment of DP, we firmly believe that investigating GNN performance under DP requirements is both valuable in theory and practice.

On the other hand, we also acknowledge that requiring node-level Differential Privacy (DP) can significantly impact the utility of GNNs or necessitate a high privacy budget to maintain reasonable utility. In fact, we are the first to highlight this limitation of DP-GNNs, as prior works (e.g., GAP) only demonstrate the case where GAP outperforms MLP on three datasets. In contrast, our extensive testing on seven datasets reveals this phenomenon, underscoring the importance of addressing this challenge. It remains an interesting and open problem to explore whether a DP-GNN design can achieve node-level DP with a moderate privacy budget while maintaining comparable utility, or whether there is a fundamental price to pay for node-level DP.

The above observations motivated us to introduce the concept of "k-neighbor-level" DP. As mentioned in our paper (lines 56-64), selecting $k = 0$ implies no privacy protection on the graph structure, while $k=n$ indicates node-level DP.  In other words, it offers a trade-off between the edge information and utility and mitigates the possibly overly pessimistic node-level DP. It is also worth noting that in many practical scenarios, edge information can be less sensitive than node features, making this notion of privacy particularly useful in various applications.

Lastly, we recognize that further research is necessary to determine whether the extent of utility drop is indeed the fundamental privacy price we must pay in the graph learning scenario. Nonetheless, exploring this aspect lies outside the scope of this single paper, but we hope that future studies will provide new answers to the challenging graph DP problem.

- G4: `` The novelty of DPDGC with respect to LINKX`` (Reviewer WwFs)

Note that we cited LINKX [38] in the original manuscript on line 198, where we also mentioned that our DPDGC is motivated by LINKX. We would like to emphasize the key novelty of DPDGC with respect to LINKX. LINKX focuses on learning with heterophilic graph datasets, and the authors purely focus on the utility aspect of the graph learning problem. In contrast, we are the first to identify and study the theoretical privacy benefit of this decoupled graph convolutional design and propose a corresponding privatization design. None of these ideas appear in the LINKX [38] paper or other prior literature. Furthermore, all prior DPGNNs leverage the “standard coupling graph convolution” design as their building block, which is significantly different from our approach. Thus, we believe that our DPDGC model is significantly novel.

---

### Decision · Program_Chairs · 2023-09-21

**Decision:**

Accept (poster)

**Comment:**

This paper was a borderline case for which the reviewers did not come to a consensus regarding the acceptance or rejection of the paper. Several reviewers also had limited confidence in their evaluation. Consequently, the Area Chair also looked closely at the paper in addition to the reviews, author response and discussions. While this work has some shortcomings, it appears to be a worthy addition to the DP-GNN literature:
- It introduces a new variant of DP for graphs which interpolates between edge DP (often considered too weak) and node DP (often too strong to yield any utility);
- It proposes a new GNN architecture well-suited to privacy, along with a clean privacy analysis.
- It provides a solid empirical evaluation showing the relevance of the proposed privacy notion and architecture, but also the limitations of DP in certain regimes.

For these reasons, I recommend acceptance.